EMBO
*reports*

# The breast cancer oncogene IKKε coordinates mitochondrial function and serine metabolism

Ruoyan Xu[1,†], William Jones[1,†], Ewa Wilcz-Villega[1], Ana SH Costa[2,3], Vinothini Rajeeve[4], Robert B Bentham[5,6], Kevin Bryson[7], Ai Nagano[1], Busra Yaman[1], Sheila Olendo Barasa[1], Yewei Wang[1], Claude Chelala[1], Pedro Cutillas[3], Gyorgy Szabadkai[5,6,8] [ID], Christian Frezza[2] & Katiuscia Bianchi[1,*] [ID]

## Abstract

IκB kinase ε (IKKε) is a key molecule at the crossroads of inflammation and cancer. Known to regulate cytokine secretion via NFκB and IRF3, the kinase is also a breast cancer oncogene, overexpressed in a variety of tumours. However, to what extent IKKε remodels cellular metabolism is currently unknown. Here, we used metabolic tracer analysis to show that IKKε orchestrates a complex metabolic reprogramming that affects mitochondrial metabolism and consequently serine biosynthesis independently of its canonical signalling role. We found that IKKε upregulates the serine biosynthesis pathway (SBP) indirectly, by limiting glucose-derived pyruvate utilisation in the TCA cycle, inhibiting oxidative phosphorylation. Inhibition of mitochondrial function induces activating transcription factor 4 (ATF4), which in turn drives upregulation of the expression of SBP genes. Importantly, pharmacological reversal of the IKKε-induced metabolic phenotype reduces proliferation of breast cancer cells. Finally, we show that in a highly proliferative set of ER negative, basal breast tumours, IKKε and PSAT1 are both overexpressed, corroborating the link between IKKε and the SBP in the clinical context.

**Keywords** ATF4; breast cancer; IKKε; mitochondrial metabolism; serine biosynthesis
**Subject Categories** Cancer; Metabolism; Signal Transduction

## Introduction

Chronic inflammation, triggered by the tumour stroma or driven by oncogenes, plays a central role in tumour pathogenesis (Netea *et al*,

2017). A key step leading to inflammation in both compartments is activation of the transcription factor nuclear factor κB (NFκB), mediated via canonical or alternative, non-canonical pathways. Key players in both pathways are the members of the IκB kinase (IKK) family, which, by phosphorylating IκB, induce its proteasome-mediated degradation, a step required for the release of NFκB from its IκB-imposed cytosolic localisation, thus leading to its nuclear translocation (Clément *et al*, 2008).

Evidence in support of the crucial role played by the IKK family in inflammation-induced malignant transformation was provided by the reduction of tumour incidence following the deletion of the canonical IKK family member IKKβ in intestinal epithelial and myeloid cells in a mouse model of colitis-associated cancer development (Greten *et al*, 2004). Soon after, the non-canonical member of the IKK family, IKKε, was shown to induce breast cancer (Boehm *et al*, 2007) and to be overexpressed in ovarian (Guo *et al*, 2009), prostate (Péant *et al*, 2011) and non-small cell lung cancers (Guo *et al*, 2013), pancreatic ductal carcinoma (Cheng *et al*, 2011) and glioma (Guan *et al*, 2011). In particular, IKKε was shown to induce breast cancer via mechanisms involving CYLD (Hutti *et al*, 2009) and TRAF2 (Zhou *et al*, 2013), ultimately mediating NFκB activation (Boehm *et al*, 2007).

Beyond cancer, IKKε is a key regulator of both innate and adaptive immunity, activating NFκB and interferon regulatory factor 3 (IRF3), inducing type I interferon signalling (Clément *et al*, 2008; Zhang *et al*, 2016), although activation of the interferon response is not essential for IKKε-mediated cellular transformation (Boehm *et al*, 2007). On the other hand, IKKε has been shown to regulate central carbon metabolism both in immune and cancer cells. In dendritic cells (DCs), IKKε, together with its closest homologue TANK binding kinase 1 (TBK1), is required for the switch to aerobic glycolysis induced by activation of the Toll-like receptors (TLRs) and activation of DCs. Glycolysis is the main glucose catabolic pathway whereby, through a series of reactions, cells metabolise glucose

1  Centre for Molecular Oncology, Barts Cancer Institute, Queen Mary University of London, John Vane Science Centre, Charterhouse Square, London, UK
2  Medical Research Council Cancer Unit, University of Cambridge, Hutchison/MRC Research Centre, Cambridge, UK
3  Cold Spring Harbor Laboratory, Cold Spring Harbor, NY, USA
4  Centre for Haemato-Oncology, Barts Cancer Institute, Queen Mary University of London, John Vane Science Centre, Charterhouse Square, London, UK
5  Department of Cell and Developmental Biology, Consortium for Mitochondrial Research, University College London, London, UK
6  Francis Crick Institute, London, UK
7  Department of Computer Sciences, University College London, London, UK
8  Department of Biomedical Sciences, University of Padua, Padua, Italy
   *Corresponding author. Tel: +44 20 7882 2049; E-mail: k.bianchi@qmul.ac.uk
   †These authors contributed equally to this work

to pyruvate which, in the presence of oxygen, is in turn oxidised to $CO_2$ in the mitochondrial matrix via the TCA cycle to produce ATP using the mitochondrial respiratory chain. Lack of oxygen prevents the mitochondrial utilisation of pyruvate, meaning glucose is instead converted into lactate (anaerobic glycolysis). In contrast, aerobic glycolysis refers to a metabolic condition whereby glucose is not fully oxidised in the mitochondria, even in the presence of oxygen, and is utilised for the production of amino acids, lipids and nucleotides via pathways branching out from glycolysis and the TCA cycle. Accordingly, aerobic glycolysis in DCs allows fatty acid synthesis, which is required for the expansion of the endoplasmic reticulum and Golgi, supporting DC activation (Everts *et al*, 2014). Allowing the production of key cellular constituents, aerobic glycolysis is most frequently observed in highly proliferative cells, such as activated immune cells and cancer cells (Andrejeva & Rathmell, 2017). In agreement, IKKε also regulates glucose uptake in pancreatic ductal adenocarcinoma and mitochondrial function in mouse embryonic fibroblasts (MEFs) (Reilly *et al*, 2013; Zubair *et al*, 2016). In addition, we have recently demonstrated that IKKε and the serine biosynthesis pathway (SBP) are important for the acquisition of malignant traits in breast epithelium exposed to macrophage conditioned medium and accordingly, expression of SBP enzymes correlates with inflammation in breast cancer (Wilcz-Villega *et al*, 2020). Our findings are in line with the known function of the SBP in cancer. Indeed, phosphoglycerate dehydrogenase (PHGDH), the first enzyme of the pathway, is amplified in breast cancer and melanoma, where it functions as an oncogene (Locasale *et al*, 2011; Possemato *et al*, 2011) and the SBP is also the target of a series of oncogenes (Amelio *et al*, 2014; Yang & Vousden, 2016). However, a comprehensive investigation of the role of IKKε as regulator of cellular metabolism in cancer has not yet been carried out. Taking an unbiased approach, we followed the fate of C13 glucose upon modulation of IKKε expression. We found that IKKε inhibits the mitochondria and indirectly controls the SBP via activation of ATF4, ultimately driving the upregulation of the SBP enzymes, in particular phosphoserine aminotransferase 1 (PSAT1). Importantly, we also demonstrate that IKKε-mediated regulation of cellular metabolism is independent of the canonical signalling pathway via NFκB/IRF3. Moreover, we have identified a subset of basal, estrogen receptor negative (ER$^-$) highly proliferative breast tumours where IKKε and PSAT1 are both overexpressed, confirming the pathophysiological role of our findings. These results identify an additional role for IKKε in breast cancer, adding regulation of cellular metabolism to the canonical oncogenic mechanisms. Thus, our data suggest a synergistic mechanism of action by which alterations of cellular metabolism and inflammation driven by the IKKε oncogene support tumour growth and proliferation.

## Results

### IKKε rewires cellular metabolism

To investigate the effect of IKKε activation on metabolism, we used two cellular model systems: (i) doxycycline-inducible Flp-In 293 HA-IKKε-expressing cells and their respective GFP-expressing controls (Flp-In 293 HA-GFP cells) and (ii) two breast cancer cell lines, T47D and MDA-MB-468, where the kinase was silenced via siRNA. HEK-

293 cells do not express endogenous IKKε, and thus, we could set its expression to a level that matched those observed in IKKε expressing breast cancer cell lines (Boehm *et al*, 2007) (Fig 1A). Liquid chromatography–mass spectrometry (LC–MS) analysis of steady-state metabolite levels revealed that induction of IKKε expression affected a large fraction of the measured metabolites (26 out of 32, Fig 1B and Dataset EV1). To account for any possible effect of doxycycline on cell metabolism, we compared cells with doxycycline-induced expression of IKKε versus GFP (Ahler *et al*, 2013). Of particular interest, IKKε increased cellular glucose and glutamine levels, along with a group of amino acids, including serine and glycine. The increased intracellular level of serine was likely a consequence of increased biosynthesis as we observed a significant increase in the level of $^{13}C_6$-glucose-derived serine (m + 3 isotopologue), suggesting that IKKε positively regulates the SBP (Fig 1C). A key enzyme in the SBP is phosphoserine aminotransferase 1 (PSAT1), which transfers nitrogen from glutamine-derived glutamate to phosphohydroxypyruvate, generating phosphoserine for the final dephosphorylation step of serine biosynthesis (Fig 1D). Using $^{15}N_2$-glutamine labelling, we confirmed increased levels of labelled serine (m + 1) in IKKε expressing cells (see Fig 1C and Dataset EV1), consistent with an increase in PSAT1 transamination activity, supporting our hypothesis that serine biosynthesis was activated by IKKε. In contrast, we observed a significant reduction in the accumulation of the TCA cycle intermediates citrate m + 2 and malate m + 2 from $^{13}C_6$-glucose, indicating that IKKε reduces pyruvate dehydrogenase (PDH) activity (Fig 1E). Fractional enrichment analysis of the above metabolites showed that in addition to increased serine biosynthesis, IKKε also augmented serine uptake from the media, shown by an increase in serine m + 0 isotopologue (Fig 1F). Moreover, the fraction of $^{13}C_6$-glucose-derived citrate and malate (m + 2, pyruvate dehydrogenase generated) was reduced in IKKε expressing cells, causing reduction in their total levels, indicating that no other carbon sources (e.g. glutamine) compensate for the lack of pyruvate entering the TCA cycle (Fig 1F). Finally, the fraction of $^{15}N$ labelled serine derived from glutamine was also increased, indicating higher glutamine usage in serine biosynthesis as nitrogen source (Fig 1G).

We then investigated whether IKKε has a similar metabolic function in breast cancer cell lines, where it is constitutively expressed. Since IKKε has been shown to be an oncogene in different breast cancer subtypes (Boehm *et al*, 2007), we used T47D and MDA-MB-468 cell lines to model estrogen receptor positive (ER$^+$) and triple-negative breast cancer, respectively (Subik *et al*, 2010). After silencing the kinase (Fig 2A and B), $^{13}C_6$-glucose and $^{15}N_2$-glutamine labelling analysis confirmed the overall effect of IKKε on cellular metabolism. In the serine and glycine biosynthesis pathways, IKKε silencing exerted the opposite effect as compared to IKKε induction in the Flp-In 293 model (Fig 2C–F and Dataset EV1). Similarly, IKKε knockdown resulted in increased levels of the TCA cycle metabolites citrate and malate m + 2 isotopologues, derived from $^{13}C$-glucose via PDH (Fig 2E and F). Taken together, these data indicated that in cancer cells IKKε redirects a significant fraction of glucose-derived carbons to the SBP and reduces pyruvate oxidation in the TCA cycle.

### IKKε inhibits mitochondrial function via PDH

Considering the effect on the TCA cycle observed via tracer compounds (see Figs 1 and 2), we assumed that IKKε alters

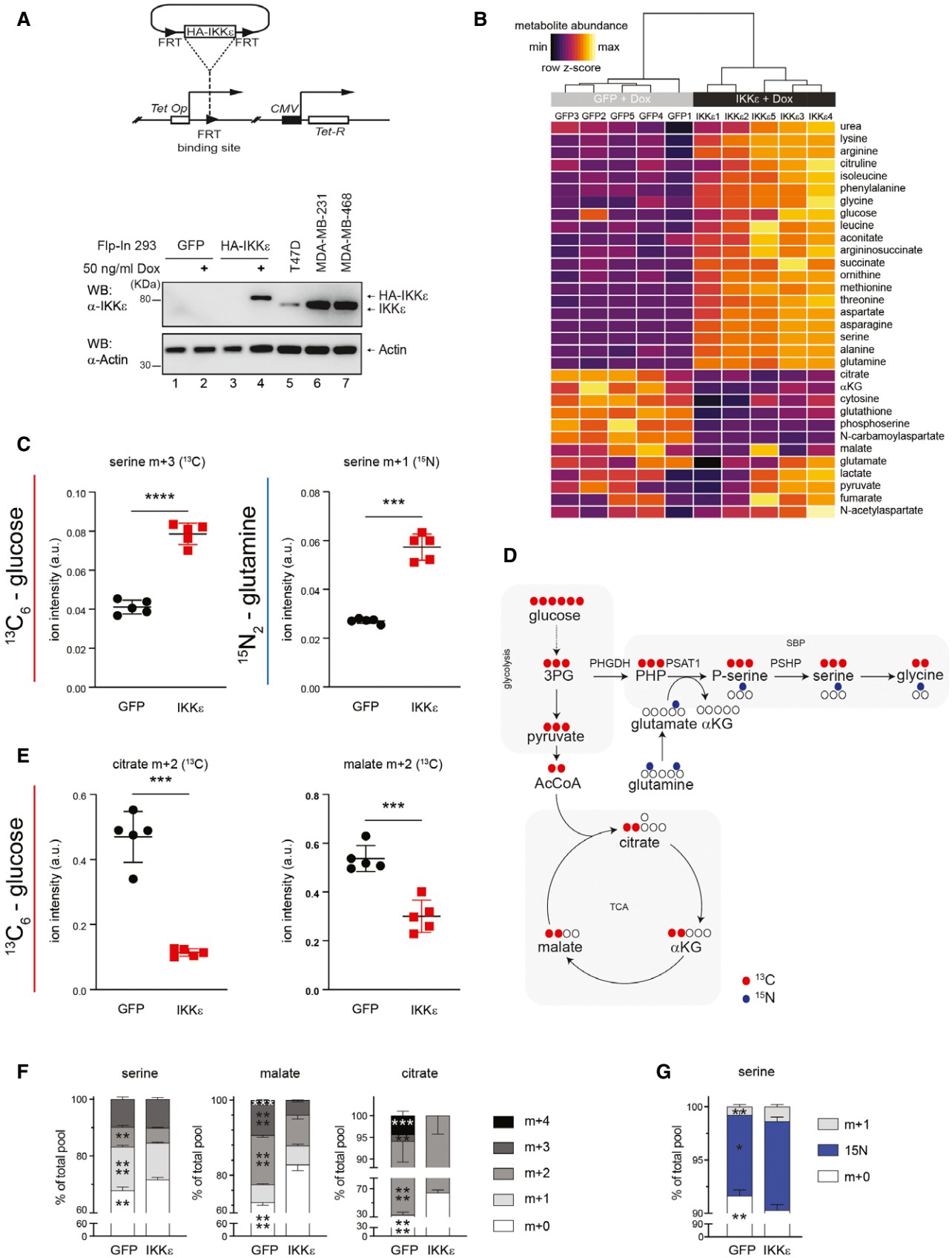

**Figure 1.**

◄

**Figure 1.  IKKε induces remodelling of cellular carbon metabolism by activating the serine biosynthesis pathway (SBP) and suppressing pyruvate oxidation.**

A   Top panel: Scheme illustrating the tetracycline-inducible Flp-In 293 system that controls the expression of HA-IKKε or HA-GFP. Bottom panel: Representative Western blot showing induced expression of HA-IKKε in Flp-In 293 cells treated with doxycycline (Dox, 50 ng/ml) for 16 h compared to endogenous IKKε in T47D, MDA-MB-231 and MDA-MB-468 breast cancer cell lines.

B   Heatmap and hierarchical clustering of metabolite concentrations in Flp-In 293 HA-GFP and Flp-In 293 HA-IKKε cells treated with doxycycline (Dox, 50 ng/ml, 16 h; $n$ = 5 technical replicates).

C   Serine production from glucose (serine m + 3, $^{13}C_6$-glucose labelling, left panel) and glutamine (serine m + 1, $^{15}N_2$-glutamine labelling, right panel) in Flp-In 293 HA-GFP or Flp-In 293 HA-IKKε cells treated with doxycycline (50 ng/ml, 16 h; $n$ = 5 technical replicates).

D   Schematic representation of the $^{13}C_6$-glucose and $^{15}N_2$-glutamine labelling strategy to assess the effect of HA-IKKε induction on glycolysis, the TCA cycle and serine metabolism.

E   Contribution of pyruvate and glucose-derived carbon to TCA cycle metabolites in Flp-In 293 HA-GFP or Flp-In 293 HA-IKKε cells treated with doxycycline (50 ng/ml, 16 h; $n$ = 5 technical replicates).

F   Fractional enrichment of serine, malate and citrate $^{13}C$-isotopologues in Flp-In 293 HA-GFP and Flp-In 293 HA-IKKε cells treated with doxycycline (50 ng/ml, 16 h; $n$ = 5 technical replicates).

G   Fractional enrichment of the serine $^{15}$-*N*-isotopologue in Flp-In 293 HA-GFP and Flp-In 293 HA-IKKε cells treated with doxycycline (50 ng/ml, 16 h). m + 1 shows the naturally occurring $^{13}C$ isotopologue ($n$ = 5 technical replicates).

Data Information: In (C, E–G), metabolite levels were normalised to the internal standard HEPES. In (C) and (E–G), data are presented as mean ± SD, *$P$ < 0.05, **$P$ < 0.01, ***$P$ < 0.001, ****$P$ < 0.0001 (two-tailed Student's $t$-test or Mann–Whitney test).

Source data are available online for this figure.

mitochondrial oxidative function. Indeed, mitochondrial oxygen consumption rate (OCR) was suppressed by IKKε induction in the Flp-In 293 HA-IKKε cell line, accompanied by reduced mitochondrial membrane potential ($\Delta\psi_m$), as assessed by respirometry and steady-state tetramethyl-rhodamine methylester (TMRM) intensity imaging. Of note, when using Flp-In 293 cells that express mutant variants of HA-IKKε, which feature mutations disrupting the function of IKKε's kinase domain (KD-m) and Ubiquitin-like domain (UbLD-m), we confirmed that both functional domains of the kinase (Ikeda *et al*, 2007) were required to exert inhibition of mitochondria (Fig 3A and B). Furthermore, IKKε silencing resulted in significantly higher OCR in a set of breast cancer cells (Figs 3C and EV1A). IKKε primarily affected ATP-coupled respiration, without significantly inhibiting uncoupled or reserve OCR, as shown by measuring respiration in the presence of oligomycin (inhibitor of the ATP synthase), and the uncoupler FCCP, respectively (Fig EV1B–F). Moreover, the effect was integral to the mitochondria, since mitochondria isolated from IKKε expressing cells showed reduced respiration compared to those isolated from GFP-expressing controls (Fig 3D).

In order to elucidate the mechanism by which IKKε regulates mitochondrial metabolism, we compared the phosphoproteomes of three independent control (GFP) and IKKε expressing Flp-In 293 clones. Multivariate analysis showed that the two clones highly

expressing IKKε grouped together in principal component analysis (PCA) and were separated from controls and cells expressing IKKε at low levels (Fig 3E and F). These results suggested that IKKε induces a dose-dependent effect in the phosphoproteome of these cells. We identified more than 3,000 phosphopeptides quantified in four technical replicates, which interestingly included the E1 subunit of the pyruvate dehydrogenase complex (PDHA1 - pS232) (Dataset EV2).

Phosphorylation of PDHA1 on S232 is known to inhibit PDH activity and is also reported to be necessary for tumour growth (Golias *et al*, 2016), and thus, we hypothesised that IKKε regulates pyruvate entry in the TCA cycle and consequently electron provision for the respiratory chain. Of note, other phosphosites of PDHA1 were either unchanged or less phosphorylated, indicating that the increase in pS232 is not due to higher level of expression of the protein (Dataset EV2). In agreement with our hypothesis, PDH activity was reduced in IKKε expressing cells (Fig 3G), and the effect was reverted by inhibiting pyruvate dehydrogenase kinase using dichloroacetic acid (DCA) (Stacpoole, 1989). DCA restored both IKKε-mediated reduction of $\Delta\psi_m$ and inhibition of respiration in Flp-In 293 mitochondria, but had no effect in control cells (Fig 3H and I), indicating that diminished pyruvate oxidation by the PDH complex is the limiting factor of respiratory activity in IKKε overexpressing

►

**Figure 2.  The effect of IKKε silencing on metabolism of breast cancer cell lines.**

A, B   Representative Western blot showing the level of IKKε in *IKBKE* (IKKε)-silenced (A) T47D and (B) MDA-MB-468 breast cancer cell lines.

C, D   Heatmap and hierarchical clustering of metabolite concentrations in (C) *IKBKE* (IKKε)-silenced T47D cells and (D) *IKBKE* (IKKε)-silenced MDA-MB-468 cells ($n$ = 5 technical replicates).

E   Glycine production, representative of serine production, from glutamine (glycine m + 1, $^{15}N_2$-glutamine labelling) and glucose (glycine m + 2, $^{13}C_6$-glucose labelling), and contribution of pyruvate and glucose-derived carbon to TCA cycle metabolites (citrate m + 2, malate m + 2, $^{13}C_6$-glucose labelling) in *IKBKE* (IKKε)-silenced T47D cells. ($n$ = 5 technical replicates).

F   Serine production from glutamine (serine m + 1, $^{15}N_2$-glutamine labelling), and serine and glycine production from glucose (glycine m + 2, serine m + 3, $^{13}C_6$-glucose labelling) as well as contribution of pyruvate and glucose-derived carbon to TCA cycle metabolites (malate m + 2, $^{13}C_6$-glucose labelling) in *IKBKE* (IKKε)-silenced MDA-MB-468 cells. ($n$ = 5 technical replicates).

Data Information: In (C, E), metabolite levels were normalised to the internal standard HEPES. In (D, F), metabolite levels were normalised to total ion count. In (C, D), metabolite levels were scaled to maximum and minimum levels of each metabolite. In (E, F), data are presented as mean ± SD, *$P$ < 0.05, **$P$ < 0.01, ***$P$ < 0.001, ****$P$ < 0.0001 (two-tailed Student's $t$-test).

Source data are available online for this figure.

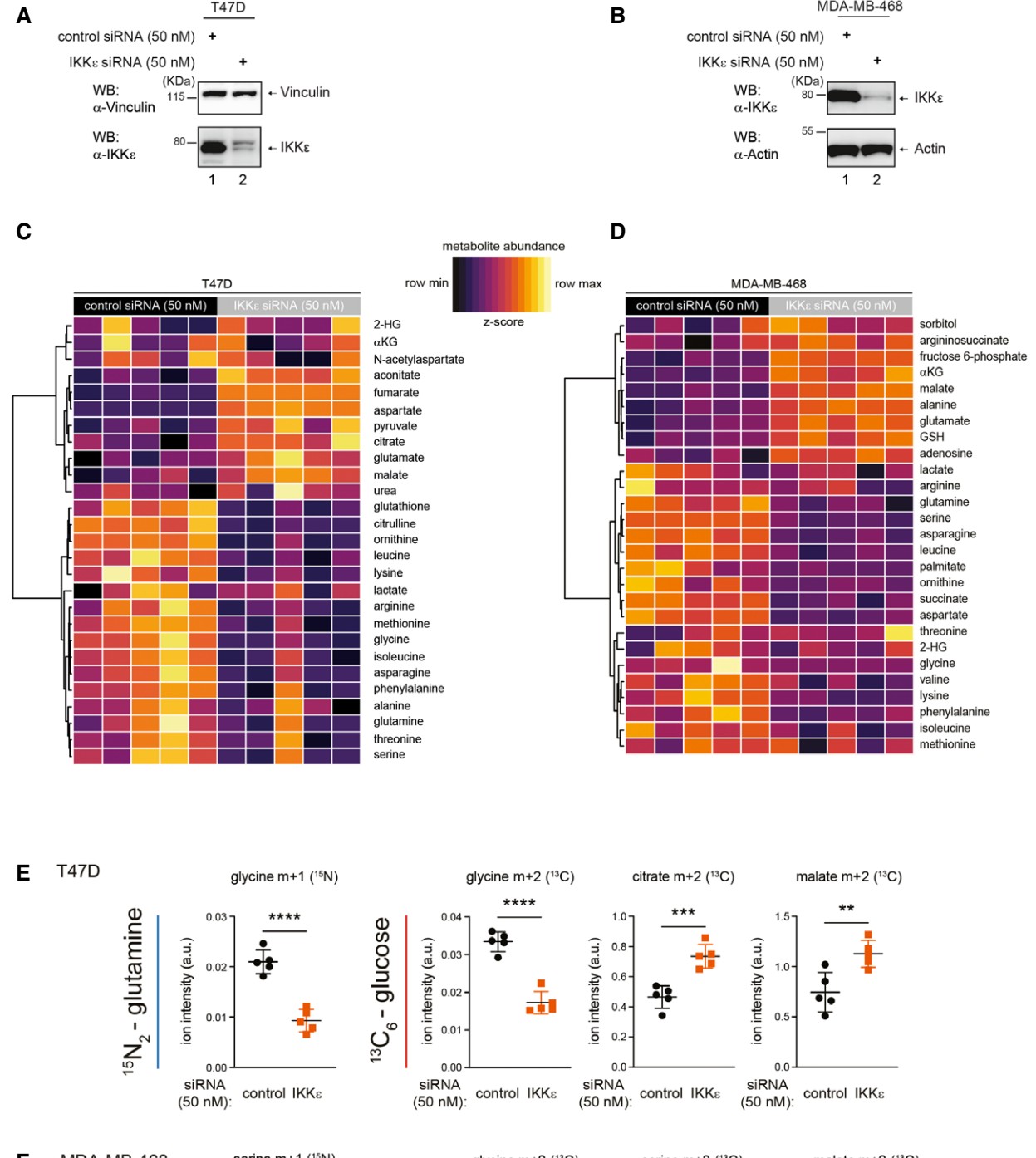

Figure 2.

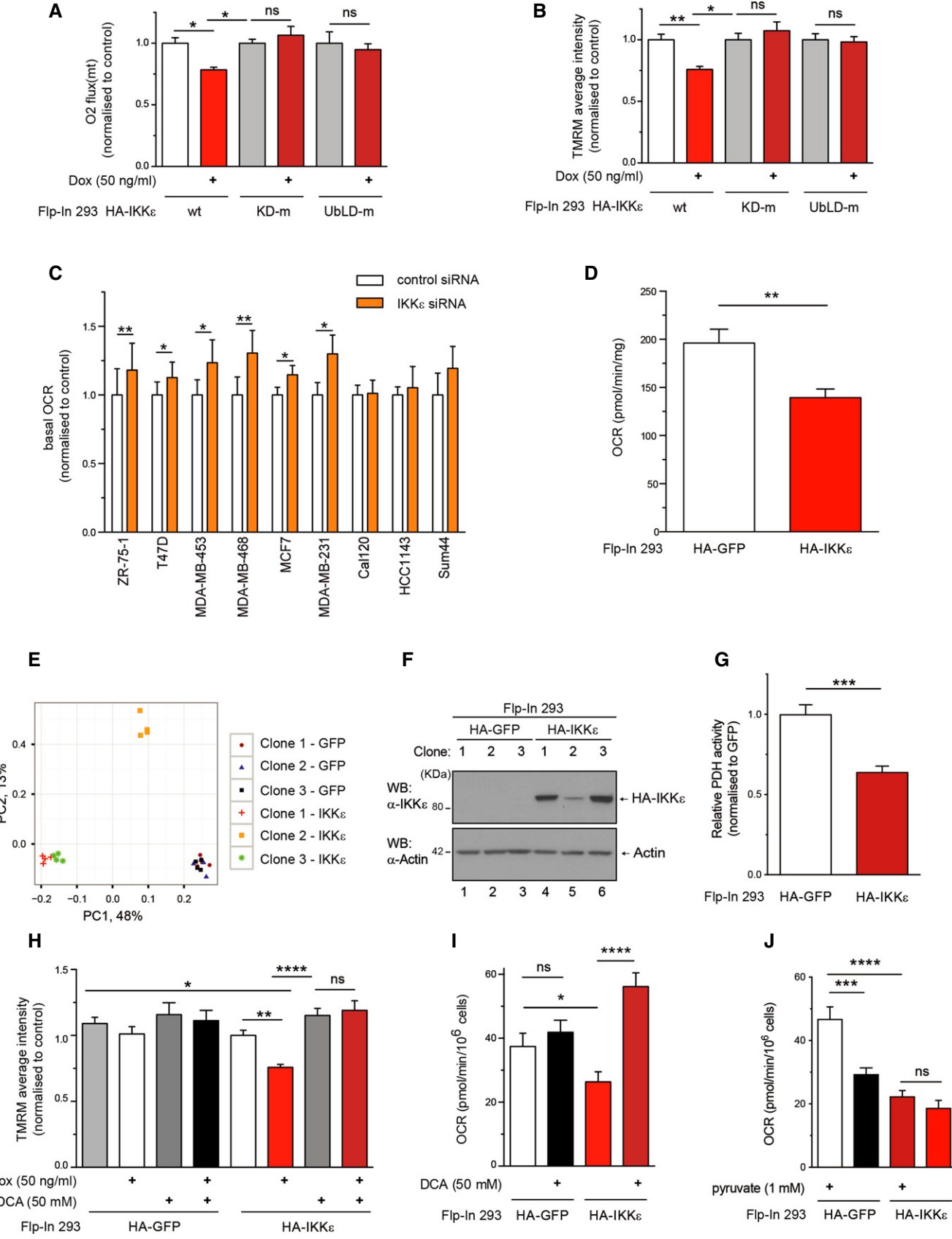

**Figure 3.**

**Figure 3.  IKKε inhibits mitochondrial metabolism via reducing PDH activity.**

A   Basal oxygen consumption in Flp-In 293 HA-IKKε wt, Flp-In 293 HA-IKKε KD-m and Flp-In 293 HA-IKKε UbLD-m cells following treatment with doxycycline (Dox, 16 h), measured using Oroboros high-resolution respirometry. Data are normalised to non-treated control cells.

B   Average TMRM staining intensity in Flp-In 293 HA-IKKε wt, Flp-In 293 HA-IKKε KD-m and Flp-In 293 HA-IKKε UbLD-m expressing cells induced by doxycycline (Dox, 16 h). Data are normalised to non-treated control cells.

C   Basal mitochondrial oxygen consumption rate (OCR) in a panel of *IKBKE* (IKKε)-silenced breast cancer cell lines, measured using Seahorse XF96e or XF24 analysis.

D   OCR in mitochondria isolated from Flp-In 293 HA-GFP or HA-IKKε cells treated with doxycycline (50 ng/ml, 16 h), measured using Oroboros high-resolution respirometry.

E   Principal component analysis of differentially phosphorylated substrates in three independent single cell clones of Flp-In 293 HA-GFP or Flp-In 293 HA-IKKε cells treated with doxycycline (100 ng/ml, 16 h). The phosphoproteomes in the three clones were analysed by mass spectrometry as described in material and methods.

F   Representative Western blot showing level of IKKε in three independent single cell clones of Flp-In 293 HA-GFP or Flp-In 293 HA-IKKε cells following treatment with doxycycline (100 ng/ml, 16 h).

G   Relative pyruvate dehydrogenase (PDH) activity in Flp-In 293 HA-GFP or Flp-In 293 HA-IKKε cells treated with doxycycline (50 ng/ml, 16 h).

H   Average TMRM staining intensity in Flp-In 293 HA-GFP or Flp-In 293 HA-IKKε cells treated with doxycycline (Dox) and dichloroacetate (DCA) (both for 16 h). Data are normalised to non-treated Flp-In 293 HA-IKKε cells.

I   Basal OCR in Flp-In 293 HA-GFP or Flp-In 293 HA-IKKε cells treated with doxycycline (50 ng/ml) in combination with DCA for 16 h, measured using Oroboros high-resolution respirometry.

J   Basal OCR in Flp-In 293 HA-GFP or Flp-In 293 HA-IKKε cells treated with doxycycline (50 ng/ml) in combination with pyruvate deprivation for 16 h, measured using Oroboros high-resolution respirometry.

Data Information: All data are $n \geq 3$ biological replicates, with the exception of (E) which is $n = 4$ technical replicates. Data are presented as mean $\pm$ SEM, *$P < 0.05$, **$P < 0.01$, ***$P < 0.001$, ****$P < 0.0001$. In (C, G), data were normalised to total sample protein concentration. The following statistical tests were applied: in (A, B) two-way ANOVA with Fisher's LSD *post hoc* tests, in (C, D, G) two-tailed paired Student's *t*-tests, in (H, I) one-way ANOVA with Fisher's LSD *post hoc* tests and in (J) one-way ANOVA with Tukey's multiple comparison tests.

Source data are available online for this figure.

cells. In line with this conclusion, we also showed that IKKε overexpressing cells rely less on pyruvate for their respiration in comparison with control cells, expressing GFP (Fig 3J).

**IKKε activates the SBP transcriptional response via ATF4**

Our tracer experiments indicated that in addition to inhibiting mitochondria, IKKε also stimulated serine biosynthesis (Figs 1 and 2). Since mitochondrial dysfunction has previously been shown to induce activation of activating transcription factor 4 (ATF4) and to regulate SBP gene transcription (Bao *et al*, 2016; Khan *et al*, 2017), we hypothesised that IKKε-induced mitochondrial inhibition elicits a similar ATF4-mediated response, in turn inducing serine biosynthesis. Confirming this hypothesis, ATF4 was induced in IKKε expressing cells (Fig 4A), while c-Myc, another known regulator of serine metabolism (Nikiforov *et al*, 2002; Sun *et al*, 2015; Anderton *et al*, 2017), remained unchanged (Fig EV2A). We therefore assessed the overall level of the three enzymes of the pathway: PHGDH, PSAT1 and phosphoserine phosphatase (PSPH). We observed that IKKε induction in Flp-In 293 HA-IKKε cells led to a marked (2–6 fold) increase in the transcription of all three SBP enzyme mRNAs (Fig 4B), which was also reflected at the protein level (with the exception of PHGDH). Of note, consistently with the role of IKKε as key mediator of the innate immune response, IRF3 was also phosphorylated upon induction of the kinase, confirming the activation of canonical kinase signalling in addition to the upregulation of ATF4 and SBP enzymes (Fig 4C). Importantly, silencing of ATF4 abolished the transcriptional upregulation of SBP enzymes mediated by IKKε and reduced their protein levels (Fig 4D and E), demonstrating the requirement of the transcription factor for the upregulation of SBP enzymes by the kinase. Moreover, while the enzymes of the SBP were upregulated in Flp-In 293 HA-IKKε cells, we did not observe differences in the level of expression of serine hydroxymethyltransferase 2 (SHMT2), the main mitochondrial enzyme involved in serine catabolism (Stover & Schirch, 1990), supporting the hypothesis that IKKε primarily

acts on serine biosynthesis. The lack of changes in SHMT2 expression is also in agreement with the lack of c-Myc involvement in the IKKε-induced pathway (Fig EV2B, and see (Nikiforov *et al*, 2002)).

Confirming the role of IKKε/ATF4 observed in our HEK model cell line, silencing of IKKε in a panel of breast cancer cell lines had the opposite effect. Upon siRNA-mediated knockdown of IKKε, downregulation of PSAT1 at the transcriptional level was observed in 5 out of 9 breast cancer cell lines tested (ZR-75-1, T47D, MDA-MB-468, Cal120 and HCC1143) (Fig 5A). Downregulation at the protein level was also observed in all cell lines where the protein was detected (with the exception of MDA-MB-231). At the protein level, we also observed downregulation of PHGDH in ZR-75-1, MDA-MB-468 and MCF7 cells and of PSPH in ZR-75-1, MDA-MB-453 and MDA-MB-468 cells (Fig 5B–E), while no increase in SHMT2 was observed (Fig EV2C and D). Moreover, reduction of the SBP enzymes was also observed in breast cancer cell lines upon the silencing of ATF4 (Fig 5F and G), in agreement with previous data (DeNicola *et al*, 2015).

Altogether, these data indicated that IKKε orchestrates a complex metabolic reprogramming that encompasses the inactivation of mitochondrial metabolism and the consequent transcriptional activation of the SBP, mediated by ATF4.

Importantly, siRNA of PSAT1 in Flp-In 293 HA-IKKε or in the breast cancer cell lines panel had no effect on oxygen consumption, further supporting that IKKε-mediated regulation of the SBP is a secondary event to regulation of the mitochondria (Fig EV3A and B).

**IKKε induces SBP gene transcription via a non-canonical mechanism**

Next, we tested the involvement of the canonical downstream signalling pathways known to be activated by the kinase. Silencing of the transcription factors IRF3 and p65 (Clément *et al*, 2008) did not abolish the induction of SBP enzyme gene transcription by

**A**

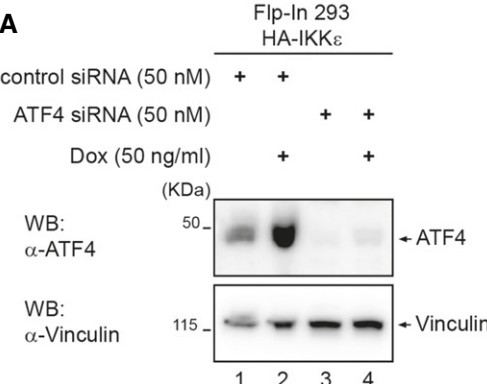

**B**

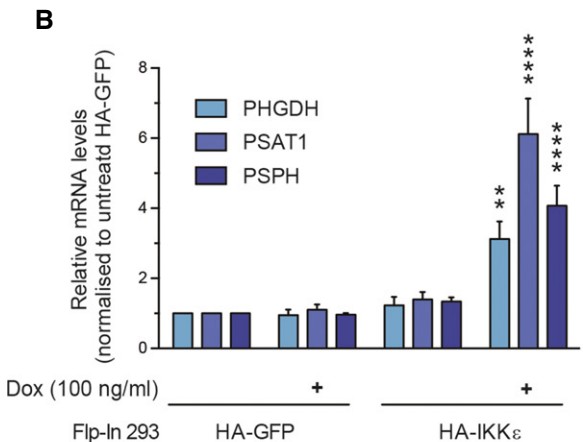

**C**

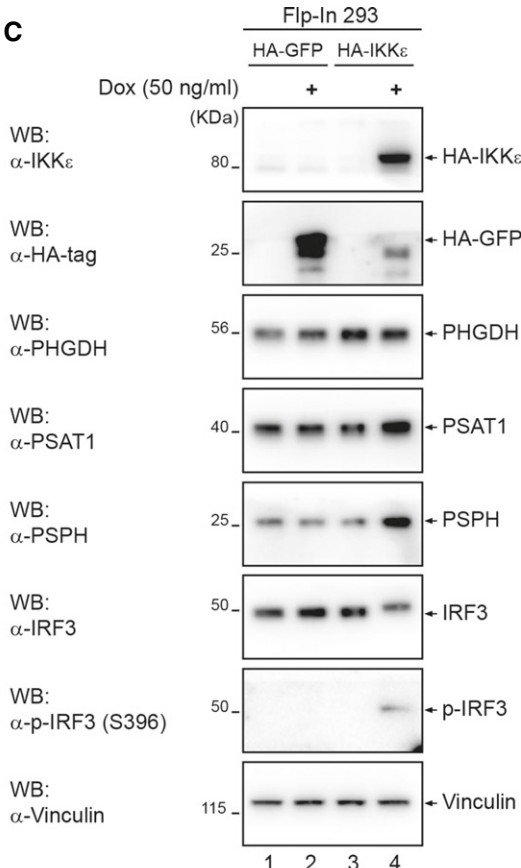

**D**

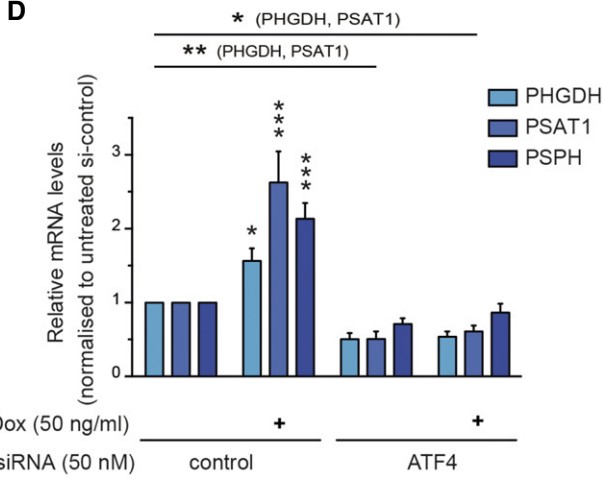

**E**

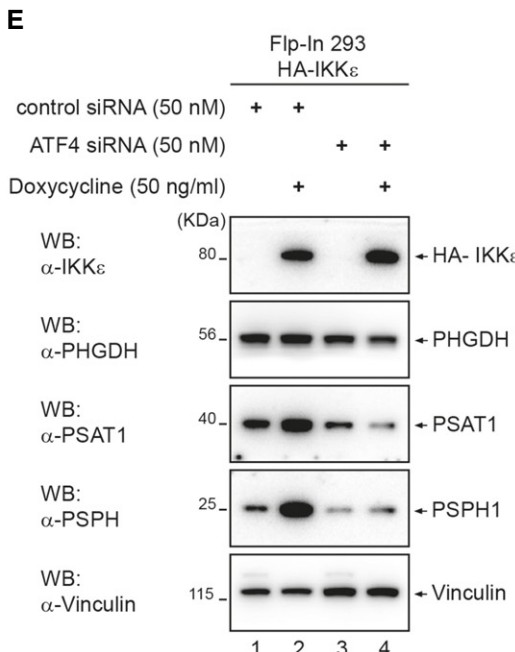

Figure 4.

**Figure 4.   IKKε stimulates SBP enzyme gene transcription via ATF4.**

A   Representative Western blot showing the level of ATF4 in *ATF4*-silenced Flp-In 293 HA-IKKε cells treated with doxycycline (Dox) for 16 h.
B   qRT–PCR analysis of *PHGDH*, *PSAT1* and *PSPH* mRNA levels in Flp-In 293 HA-GFP or Flp-In 293 HA-IKKε cells treated with doxycycline (Dox) for 16 h. Data are expressed as fold changes, relative to levels in non-treated Flp-In 293 HA-GFP cells and normalised to *β-Actin* (n = 4 biological replicates)
C   Representative Western blot showing levels of IKKε, IRF3, phosphorylated IRF3 (S396) and SBP enzymes in Flp-In 293 HA-GFP or Flp-In 293 HA-IKKε cells treated with doxycycline (Dox) for 16 h.
D   qRT–PCR analysis of *PHGDH*, *PSAT1* and *PSPH* mRNA levels in *ATF4*-silenced Flp-In 293 HA-IKKε cells treated with doxycycline (Dox) for 16 h. Data are expressed as fold changes, relative to levels in a non-silenced, non-treated control and normalised to *β-Actin* (n = 5 biological replicates).
E   Representative Western blot showing the levels of IKKε, PHGDH, PSAT1 and PSPH in *ATF4*-silenced Flp-In 293 HA-IKKε cells treated with doxycycline (Dox) for 16 h.

Data Information: In (B, D), data are presented as mean ± SEM, statistics were performed using log-transformed fold change values. *$P < 0.05$, **$P < 0.01$, ***$P < 0.001$, ****$P < 0.0001$, measured using two-way ANOVA with Bonferroni *post hoc* tests.
Source data are available online for this figure.

IKKε (Fig EV3C–E). Of note, these experiments indicated that IRF3 is required to maintain the transcription of PHGDH in basal conditions, but has no role in IKKε-mediated induction. Finally, since IKKε induces the secretion of a range of cytokines (Barbie *et al*, 2014), we tested the possibility that the induction of SBP enzyme gene transcription is mediated by an autocrine loop. Extracellular media conditioned by HA-GFP or HA-IKKε expressing Flp-In 293 cells was collected and applied on three different receiving cell lines: Flp-In 293 HA-GFP, not expressing IKKε (Fig EV4A) and the T47D and ZR-75-1 breast cancer cell lines, constitutively expressing IKKε (Fig EV4B and C). Media conditioned by IKKε expressing cells had no effect on the SBP enzymes, even though cytokine-mediated JAK-STAT signalling was observed in all three receiving cell lines, as demonstrated by induction of STAT1, phospho-STAT1 and OAS1 (an interferon-inducible gene, only in ZR-75-1).

Altogether, these results demonstrated that IKKε induces SBP enzyme gene transcription by a cell-autonomous mechanism which, however, does not include its canonical downstream targets.

**Pharmacological inhibition of IKKε-induced metabolic changes reduces cell proliferation**

To test the functional consequences of the IKKε-mediated metabolic rewiring on tumour proliferation, we assessed the outcome of inhibiting the two key metabolic reactions of serine biosynthesis on cell proliferation in a panel of breast cancer cell lines. Significantly reduced breast cancer cell proliferation was observed upon treatment with NCT502, a recently described inhibitor of the SBP (Pacold *et al*, 2016), in four out of the eight cell lines tested (ZR-75-1, T47D, MDA-MB-468 and HCC1143), while its PHGDH inactive form only had an effect in T47D cells (Fig EV5A). Importantly, the effect on proliferation was significantly correlated with the effect of IKKε on mitochondrial OCR, but not extracellular acidification rate (Figs 6A and EV5B). These results suggested that inhibition of the specific metabolic effect regulated by IKKε correlates with cancer cell proliferation rate. The same effect was observed upon treatment with the glutamine antagonist 6-diazo-5-oxo-l-norleucine (DON) (Cervantes-Madrid *et al*, 2015) and CB839 (Gross *et al*, 2014), to inhibit glutaminase and thus glutamate availability for PSAT1 (Figs 6B and EV5C–E). These results suggest that IKKε-induced metabolic changes promote cell proliferation and might be valuable targets to inhibit IKKε-driven tumorigenesis.

**IKKε and PSAT1 are overexpressed in a common, highly proliferative subset of breast cancer**

In order to explore IKKε and SBP enzyme gene expression status in breast cancer patient samples, we analysed the METABRIC dataset (Curtis *et al*, 2012), which includes data from 1981 breast cancer patients with pathological and clinical details. *IKBKE* and *PSAT1* mRNA were significantly upregulated (above the 95% confidence interval) in 200 (10.1%) and 664 (33.5%) samples, respectively, and 107 (5.4%) samples showed overexpression of both mRNAs. This indicated a highly significant association between the two genes, as confirmed chi-square independence test (Fig 6C). Given that breast cancer is a heterogenous disease, commonly classified into 5 to 10 intrinsic subtypes (Perou *et al*, 2000; Curtis *et al*, 2012), the association of *IKBKE* and *PSAT1* might be driven by subtype-specific expression. Thus, in order to identify subtypes with significant overexpression compared to the total population, we compared the expression values of *IKBKE* and the SBP genes in all Pam50 subtypes (Parker *et al*, 2009; Jiang *et al*, 2016) and in the estrogen receptor (ER) positive and negative populations. This analysis indicated that both *IKBKE* and *PSAT1*, similarly to *PHGDH* and *PSPH*, are significantly upregulated in an ER-negative Pam50:basal subpopulation of tumours, with the highest proliferation index (Nielsen *et al*, 2010) (Fig 6D–H). ATF4-driven overexpression of *PSAT1* in ER-negative tumours has been previously shown in a different dataset (Gao *et al*, 2017) and here we also have found strong association of these two genes (Fig 6I and J). Importantly, while *PSAT1* is overexpressed in almost all ER-negative samples (378 out of 435), only 79 samples overexpress *IKBKE*, indicating that *PSAT1* is regulated by multiple inputs. However, above 90% of samples overexpressing *IKBKE* (72 out of 79) also overexpress *PSAT1*. Due to the large fraction of PSAT1 overexpressing samples in the ER-negative subset and the overall low expression and detected variability of *IKBKE* and *ATF4* mRNAs in the dataset, only low but still statistically significant levels of correlation could be found between these genes (Fig 6K and L). However, these gene expression patterns clearly show that IKKε-mediated expression of PSAT1 and the SBP enzymes, demonstrated in our *in vitro* experiments, is potentially also functional in a subset of clinical samples, suggesting the pathophysiological importance of the pathway in breast cancer. Importantly, we also confirmed the correlation between IKKε and PSAT1 expression in a set of breast cancer cases (Wilcz-Villega *et al*, 2020).

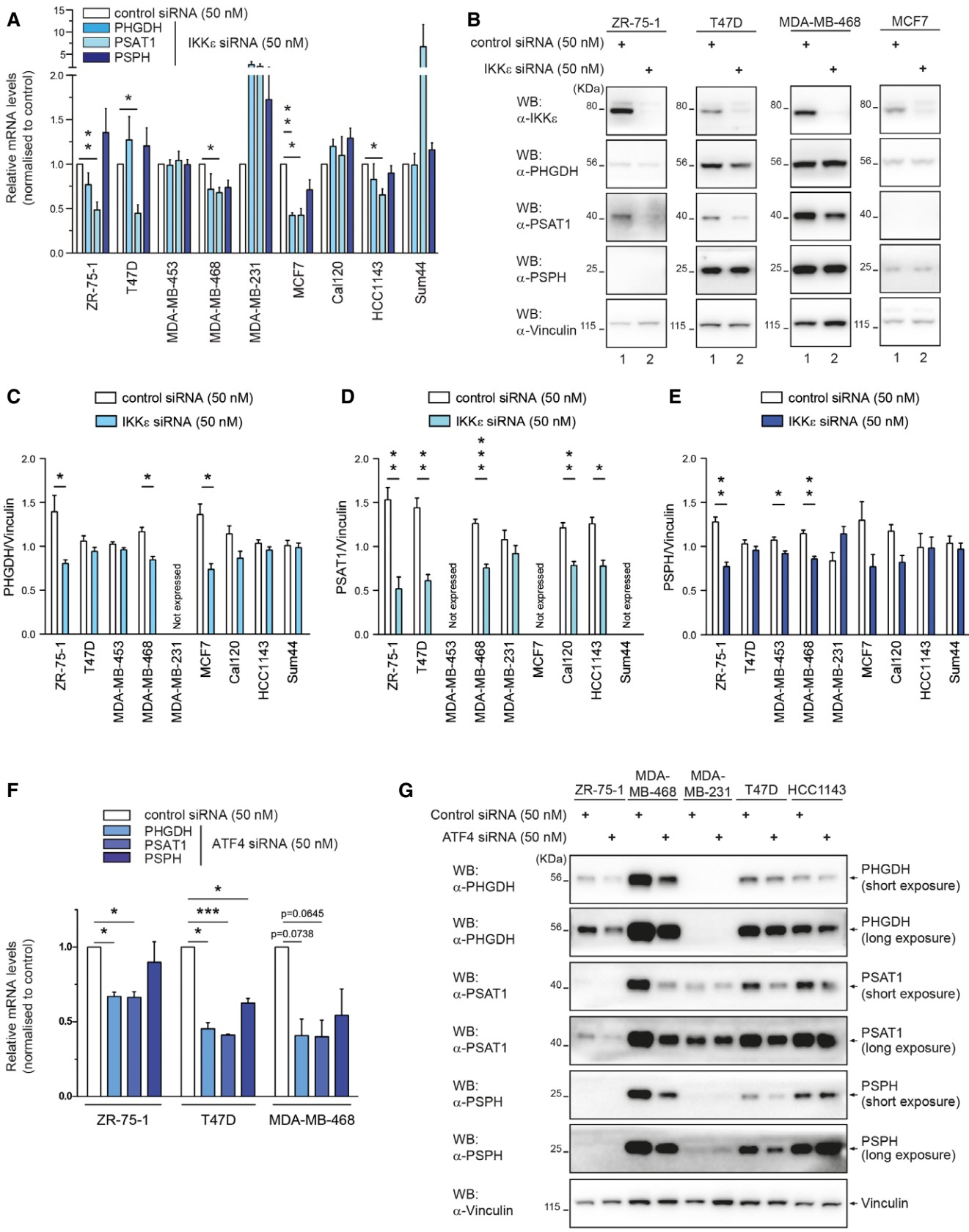

Figure 5.

**Figure 5.  The SBP is primarily regulated by an IKKε-mediated transcriptional response.**

A    qRT–PCR analysis of *PHGDH*, *PSAT1* and *PSPH* mRNA levels in a panel of *IKBKE* (IKKε)-silenced breast cancer cell lines. Data are expressed as fold changes, relative to levels in a non-silenced control of each cell line and normalised to *β-Actin* (n ≥ 3 biological replicates).

B    Representative Western blot showing the levels of IKKε and the SBP enzymes in *IKBKE* (IKKε)-silenced ZR-75-1, T47D, MDA-MB-468 and MCF7 breast cancer cell lines.

C–E   Levels of the SBP enzymes in a panel of *IKBKE* (IKKε)-silenced breast cancer cell lines. (C) PHGDH, (D) PSAT1 and (E) PSPH levels in indicated cell lines normalised to Vinculin. Densitometry analysis quantified single sample density as a percentage of total blot density per cell line prior to vinculin normalisation (n ≥ 3 biological replicates).

F    qRT–PCR analysis of *PHGDH*, *PSAT1* and *PSPH* mRNA levels in *ATF4*-silenced ZR-75-1, T47D and MDA-MB-468 breast cancer cell lines. Data are expressed as fold changes, relative to levels in non-silenced control cells and normalised to *β-Actin* (n = 3 biological replicates).

G    Representative Western blot showing the levels of PHGDH, PSAT1 and PSPH in *ATF4*-silenced ZR-75-1, MDA-MB-468, MDA-MB-231, T47D and HCC1143 breast cancer cell lines.

Data Information: In (A, C–F), data are presented as mean ± SEM. *$P < 0.05$, **$P < 0.01$, ***$P < 0.001$. In (A, F), one-sample *t*-tests were performed using log-transformed fold change values for all samples, except Sum44 PHGDH in (A), in which case a Wilcoxon signed rank was performed using log-transformed fold change values. In (C–E), two-tailed paired *t*-tests were performed.

Source data are available online for this figure.

## Discussion

Here, we described a novel fundamental mechanism by which IKKε, a key player in the innate immune response, regulates cellular metabolism. We show that the kinase orchestrates a complex metabolic reprogramming culminating in the regulation of the serine biosynthesis pathway. The mechanism involved in IKKε-mediated regulation of the SBP is inhibition of carbon supply to the mitochondria, leading to the transcriptional upregulation of SBP genes via a mitochondrial-nuclear retrograde signalling pathway targeting ATF4, ultimately activating serine biosynthesis. The overall metabolic switch induced by IKKε supports cancer cell proliferation and is present in a subset of breast tumours, providing potentially important pharmacological targets. The pathway described here is reminiscent of recent data showing that the uptake of pyruvate in mitochondria regulates the SBP (Baksh *et al*, 2020).

While such mechanistic details of the function of IKKε as a newly identified modulator of the SBP and mitochondria have not been reported before, previous studies implicated IKKε in the regulation of cellular metabolism. Consistent with our data, IKKε was shown to inhibit OCR in MEFs (Reilly *et al*, 2013) and regulate glycolysis in DCs, although in this system the kinase did not affect mitochondrial metabolism (Everts *et al*, 2014). Similarly, in pancreatic ductal adenocarcinoma, IKKε was shown to stimulate glucose uptake, but did not inhibit mitochondrial respiration (Zubair *et al*, 2016). Thus, IKKε appears to modulate cellular metabolism in a tissue- and

context-specific manner, and our study pinpoints and extends the breadth of the specific cellular targets utilised by the kinase to exert these heterogeneous responses. Importantly, in addition to the previously known canonical NFκB and IRF3 signalling pathways (Clément *et al*, 2008), IKKε can engage the mitochondrial-nuclear ATF4-mediated signalling. Whether NRF2, previously demonstrated to regulate the SBP upstream of ATF4 (DeNicola *et al*, 2015), is also involved in the signalling induced by IKKε, remains to be tested. While here we have shown that in breast cancer cells IKKε-mediated changes in metabolism support proliferation, these metabolic alterations might also facilitate other cellular functions (Jones & Bianchi, 2015), for example, cytokine secretion in immune cells (Chang *et al*, 2013; Tannahill *et al*, 2013; Rodriguez *et al*, 2019; Yu *et al*, 2019). Apart from providing novel mechanistic details of IKKε-mediated cellular metabolic changes, this work also indicates the necessity of further research to better understand the physiological and pathological role of IKKε in order to efficiently and selectively target tumour cells relying on this oncogene. Our observations suggest that drugs targeting IKKε-regulated metabolic pathways can specifically target breast cancer cells without affecting other cell types, considering that it is only in these cells that IKKε has been reported to regulate the SBP. Indeed, our gene expression analysis indicated that the IKKε-mediated pathway defines a subset of ER⁻, basal breast tumours, and thus, evaluation of the IKKε-mediated metabolic and gene expression phenotype can help to further stratify breast cancer for treatment. Our stratification is also in agreement with previous

**Figure 6.  The pathophysiological role of IKKε-induced metabolic and gene expression alterations in breast cancer.**    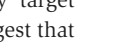

A    Correlation of change in OCR (ΔOCR) in a panel of *IKBKE* (IKKε)-silenced breast cancer cell lines (from Fig 3C) and the change in cell confluency (Δconfluency) upon treatment of the panel of cell lines with NCT502 (from Fig EV5A).

B    Correlation of ΔOCR in a panel of *IKBKE* (IKKε)-silenced breast cancer cell lines (from Fig 3C) and the Δconfluency upon treatment of the panel of cell lines with 6-Diazo-5-oxo-ʟ-norleucine (DON) (from Fig EV5C).

C    Association between *IKBKE* (IKKε) and *PSAT1* mRNA overexpression evaluated by a chi-squared independence test. The + sign indicates samples with significant (*P* < 0.05) overexpression of *IKBKE* or *PSAT1*. Number and percentage of samples, as well as the chi-square values are shown.

D–I   Expression of *IKBKE* (IKKε) and the SBP enzymes *PSAT1*, *PHGDH* and *PSPH* in the METABRIC dataset. The expression of a proliferation-related gene set and *ATF4* is also shown. Samples were stratified by Pam50 intrinsic subtypes and ER status. Brown–Forsythe and Welch ANOVA test with unpaired *t* with Welch correction were applied. *$P < 0.05$, **$P < 0.01$, ***$P < 0.001$, ****$P < 0.0001$.

J    Association between *ATF4* and *PSAT1* mRNA overexpression evaluated by chi-squared independence test. The + sign indicates samples with significant (*P* < 0.05) overexpression of *ATF4* or *PSAT1*. Number and percentage of samples, as well as the chi-square values are shown.

K, L   Correlation of *IKBKE* (IKKε) and *PSAT1* (K) or *ATF4* and *PSAT1* (L) expression in the ER-negative sample subset.

Data Information: In (A, B), cell confluency was measured using the IncuCyte Zoom, ΔOCR was measured using Seahorse XF96e or XF24 analysis. Data are n ≥ 3 biological replicates. In (A, B, K, L) linear regression, correlation coefficients (Pearson's correlation, Spearman's rho) significance of difference from slope = 0 are shown.

Source data are available online for this figure.

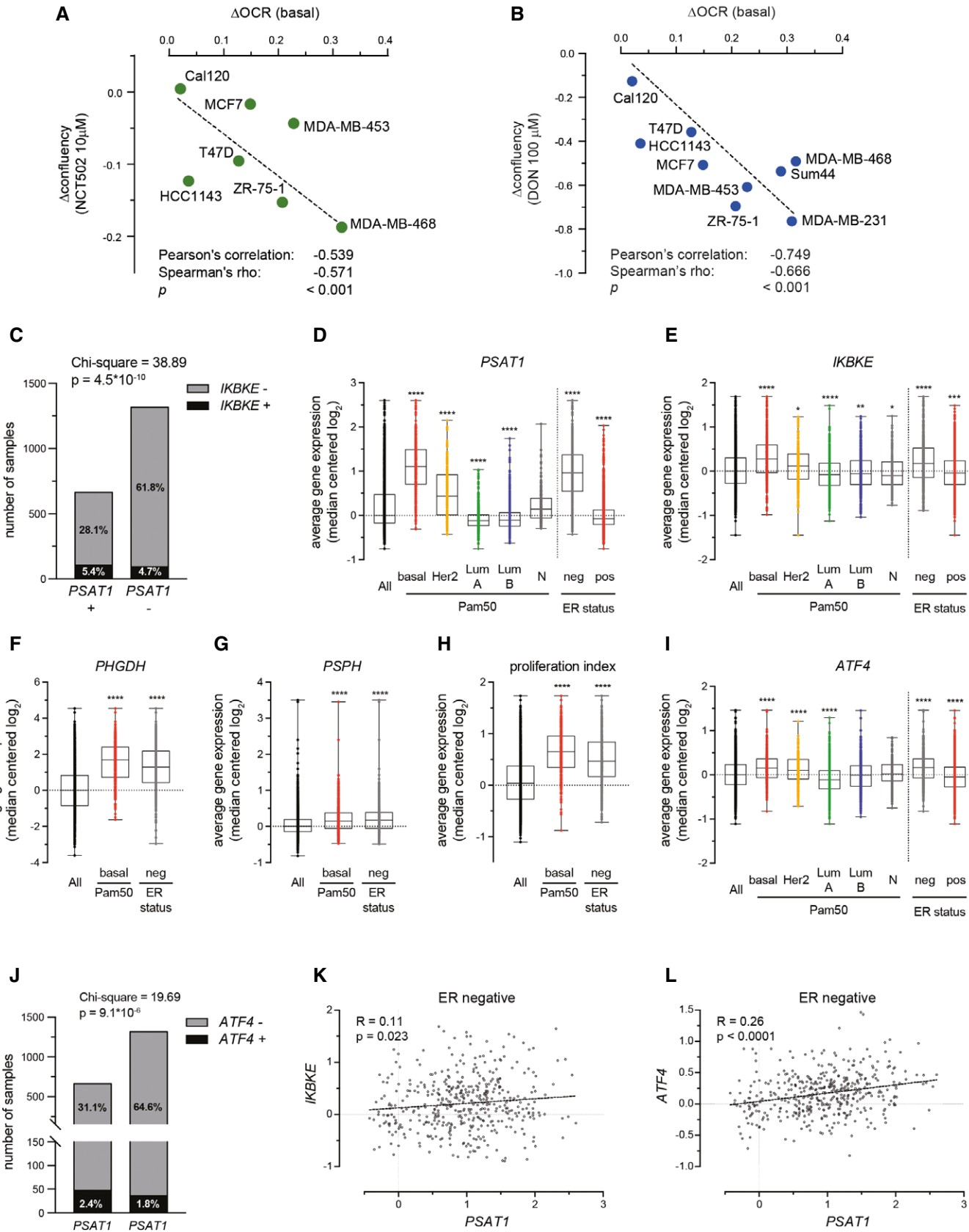

Figure 6.

studies, where strong correlation between *PSAT1* expression and tumour proliferation has been found in ER⁻ tumours (Coloff *et al*, 2016; Gao *et al*, 2017). Of note, IKKε, along with the JAK/STAT pathway, has been reported to regulate a cytokine network promoting cellular proliferation in a subset of triple-negative breast tumours (Barbie *et al*, 2014), and PSAT1 overexpression is also a feature of a small fraction of ER⁺ tumours along with the JAK-STAT pathway (De Marchi *et al*, 2017). While we have shown that IKKε activates the JAK-STAT pathway (Fig EV4) along with PSAT1 overexpression, JAK-STAT activation *per se* was not sufficient for the induction of the enzyme. This indicates that both IKKε and PSAT1 are defining features in order to identify tumours where the pathway is actively promoting proliferation.

Finally, further work is required to investigate whether the IKKε-mediated metabolic phenotype described here, especially regarding the regulation of the SBP, occurs in different cellular systems where IKKε is known to be activated. This could help to develop new therapeutic strategies applicable in a broad range of inflammation-related diseases, beyond cancer.

# Materials and Methods

## Plasmids

DNA fragments encoding wild-type human IKKε (UniProt: Q14164) were amplified separately by PCR using primers containing Kpn1 (5′) and EcoR1 (3′) restriction sites.
For 5′-ttggtaccagccagctcagggcaggagatgcagagcacagccaatta-3′
Rev 5′-gatggatatctgcagaattcaggaggtgctgggactc-3′

The PCR products were double digested by these two enzymes and ligated to vector *pcDNA5.5* (a kind gift from Dr Tencho Tenev), which provides a 2xHA tag at the c-terminal, to generate the *pcDNA5.5-wt-IKKε* plasmid. For mutant variants of IKKε with disrupted functional domains, the kinase domain mutant (KD-m) was created by site-directed mutagenesis, using primers to introduce a K38A mutation to the wild-type IKKε sequence.
For 5′-gagctggttgctgtggcggtcttcaacactac-3′
Rev 5′-gtagtgttgaagaccgccacagcaaccagc-3′

*UbLD-M-IKKε* plasmid, encoding the ubiquitin-like domain mutant (UbLD-m) variant of IKKε (containing L353A and F354A mutations) was a kind gift from Prof Ivan Dikic. Both KD-m and UbLD-m IKKε variants were amplified and ligated into the *pcDNA5.5* vector using the same Kpn1 and EcoR1 double digestion and ligation method as the wild-type kinase, generating the *pcDNA5.5-KD-M-IKKε* and *pcDNA5.5-UbLD-M-IKKε* plasmids.

## Cells

To generate Flp-In 293 cells expressing either wild type, kinase domain mutant (KD-m) or ubiquitin-like domain mutant (UbLD-m) IKKε, Flp-In 293 cells (Invitrogen) were transfected with either *pcDNA5.5-wt-IKKε*, *pcDNA5.5-KD-M-IKKε* (K38A), *pcDNA5.5-UbLD-M-IKKε* (L353A F354A) or *pcDNA5.5-GFP*, together with a pOG44 plasmid at a molar ratio of 1:9. cDNA plasmids were mixed with Lipofectamine LTX (15338100, Thermo Fisher Scientific) or Fugene HD (E2311, Promega) according to the manufacturer's instruction and transfected into the different cell lines for 48 h.

Stable cell lines and single cell clones expressing wild-type IKKε (wt) or mutant IKKε, with disruption of either kinase domain or ubiquitin-like domain function (KD-m or UbLD-m), in a doxycycline-dependent manner were selected with 300 μg/ml hygromycin (Calbiochem). All Flp-In 293 cells were cultured in DMEM (Sigma-Aldrich). The panel of breast cancer cell lines were kindly provided by Dr. Alice Shia and Prof. Peter Schmid. MDA-MB-231, MDA-MB-468, MDA-MB-175, ZR75.1, T47D, HCC1143, MCF7 were cultured in RPMI-1640 (Sigma-Aldrich), Cal120 and MDA-MB-453 were cultured in DMEM (Sigma-Aldrich) and Sum44 in DMEM (Sigma-Aldrich) and 1 nM estrogen (Sigma-Aldrich). For all cell lines, medium was supplemented with 10% FBS, penicillin–streptomycin and Normocin (InvivoGeN). Serine-free medium was custom made DMEM without serine, with 10% dialysed FBS and penicillin–streptomycin. All cells were cultured with environmental conditions of 37°C, 5% $CO_2$.

## Drugs

The following drugs were used: 6-Diazo-5-oxo-L-norleucine (Don, D2141, Sigma-Aldrich); Sodium dichloroacetate (DCA, 347795, Sigma-Aldrich); NCT-502 and PHGDH inactive (19716 and 19717, Cayman); Doxycycline (Dox, D9891, Sigma-Aldrich), Oligomycin (sc-203342, Santa Cruz Biotechnology); FCCP (sc-203578, Santa Cruz Biotechnology); Antimycin (sc-202467, Santa Cruz Biotechnology); Rotenone (sc-203242, Santa Cruz Biotechnology); Cyt.C (C2037, Sigma-Aldrich) CB-839 (10-4556, Focus Biomolecules); Adenosine diphosphate (ADP, A2754, Sigma-Aldrich).

## siRNA transfection

The following oligos were transfected for siRNA-mediated knockdown: AllStars Negative Control siRNA (1027281, Qiagen); Hs_ATF4_9 FlexiTube siRNA (SI04236337, Qiagen); Hs_IKBKE_6 FlexiTube siRNA, (S102622319, Qiagen); Hs_IKBKE_7 FlexiTube siRNA (S102622326, Qiagen); Hs_IKBKE_8 FlexiTube siRNA (S102655317, Qiagen); Hs_IKBKE_9 FlexiTube siRNA (s102655324, Qiagen); Hs_IRF3_4 FlexiTube siRNA (SI02626526, Qiagen); Hs_PSAT1_10 FlexiTube siRNA (SI03019709, Qiagen, UK); Hs_PSAT1_12 FlexiTube siRNA (SI03222142, Qiagen, UK); Hs_PSAT1_14 FlexiTube siRNA (SI04265625, Qiagen, UK); Hs_PSAT1_15 FlexiTube siRNA (SI04272212, Qiagen, UK); Hs_RELA_5 FlexiTube siRNA (SI00301672, Qiagen).

For transfection, siRNA was mixed with Dharmafect 4 (T200402, Dharmacon), and cells were transfected according to the transfection reagent manufacturer's instruction for 48 h or 72 h prior to measurements. Cells were transfected with a final concentration of 50 nM siRNA, and a pool of all 4 IKBKE-targeting oligos was used for suppression of IKKε, a pool of all 4 PSAT1-targeting oligos was used for suppression of PSAT1, and single targeting oligos were used for the suppression of ATF4, p65 and IRF3.

## Oxygen consumption and extracellular acidification rate measurements

An XF24 Extracellular or XF96e Extracellular Flux analyser (Seahorse Biosciences, Agilent Technologies) was used to determine the bioenergetic profiles in breast cancer cell lines. Cells were plated in six-well corning dishes first and then transfected with siRNA 24 h

after plating. Twenty-four hours after transfection, cells were trypsinised, counted and plated into a 24 or 96-well Seahorse plate. Oxygen consumption rates (OCR) and extracellular acidification rates (ECAR) were assessed in Seahorse medium according to the manufacturer protocols. Respiratory parameters were assessed as described in Fig EV1B. Oxygen consumption rate (OCR) of Flp-In 293 cells was measured using an Oroboros high-resolution respirometer (Oroboros) at 37°C, in Seahorse XF assay medium containing 4.5 g/l glucose, 1 mM pyruvate and 25 mM Hepes, and the assay was performed as in Fig EV1B.

For measurements in isolated mitochondria, Flp-In 293 cells were first washed with PBS and collected in homogenisation buffer (250 mM sucrose, 5 mM Hepes, pH 7.4, 0.5 mM EGTA), and Protease inhibitor cocktail (1187358001, Roche) and then homogenised in a glass/glass, tight potter by 100 strokes on ice, followed by centrifugation for 5 min at 800 $g$ at 4°C. The supernatant, containing mitochondria, was centrifuged again at 9,000 $g$. The pellet was resuspended and adjusted to a protein concentration of 0.8 mg/ml in OCB buffer (125 mM KCl, 20 mM MOPS, 10 mM Tris ph7.2–7.3, 0.2 mM EGTA, 2.5 mM $KH_2PO_4$, 2.5 mM $MgCl_2$). 10 mM glutamate and 5 mM malate were added to the mitochondrial suspension before the experiment, and OCR was measured in OCB buffer using the Oroboros high-resolution respirometer. ADP (final concentration 0.25 mM), Cyt.C (10 μM), oligomycin (5 μM) were injected step by step, and 50 μM FCCP was added in 1 μl steps until maximum respiratory capacity was detected. At the end of the run, antimycin (5 μM final concentration) was injected. Data were then analysed by the Datalab 5.5 (Oroboros) software.

## Cell proliferation assay

Cells were plated in Corning 96-well plates at a density between 2,000 and 10,000 cells per well for different cell lines. Cell proliferation rate was then measured using the IncuCyte ZOOM instrument (Essen Biosciences) for 3–7 days, and proliferation rate was analysed with the Incucyte Zoom 2015A software (Essen Biosciences).

## Metabolic labelling and metabolome analysis

Flp-In 293 cells and breast cancer cell lines (T47D and MDA-MB-468) were first plated separately in six-well plates in five technical replicas per each condition. IKKε expression in Flp-In 293 cells was then induced by 50 ng/ml doxycycline, and breast cancer cells were transfected with siRNA to suppress IKKε. Two hours after induction for the Flp-In 293 cells, and 48 h after siRNA transfection for the breast cancer cell lines, cells were incubated with either $^{13}C_6$-glucose (CLM-1396-5, Cambridge Isotope Laboratories) medium or $^{15}N_2$-glutamine (NLM-1328-0.25, Cambridge Isotope Laboratories) medium for 14 h. Cells were then washed three times with PBS, and metabolites were extracted using cold extraction buffer (50% methanol, 30% acetonitrile, 20% ultrapure water, 100 ng/ml HEPES) at a ratio of 1 ml extraction buffer/$10^6$ cells. After 15-min incubation time on methanol and dry ice, cells were placed on a shaker for 15 min using a thermal mixer at 4°C and incubated for 1 h at −20°C. Cell lysates were centrifuged, and the supernatant was collected and transferred into autosampler glass vials, which were stored at −80°C until further analysis.

Samples were randomised in order to avoid bias due to machine drift and processed blindly. LC–MS analysis was performed using a Q Exactive Hybrid Quadrupole-Orbitrap mass spectrometer coupled to a Dionex U3000 UHPLC system (Thermo Fisher Scientific). The liquid chromatography system was fitted with a Sequant ZIC-pHILIC column (150 mm × 2.1 mm) and guard column (20 mm × 2.1 mm) from Merck Millipore and temperature maintained at 45°C. The mobile phase was composed of 20 mM ammonium carbonate and 0.1% ammonium hydroxide in water (solvent A) and acetonitrile (solvent B). The flow rate was set at 200 μl/min with the gradient described previously (Mackay *et al*, 2015). The mass spectrometer was operated in full MS and polarity switching mode. The acquired spectra were analysed using Xcalibur Qual Browser and Xcalibur Quan Browser software (Thermo Fisher Scientific).

## Phosphoproteomics

### Sample preparation

Flp-In 293 single cell clones for IKKε (Clones 1,2 and 3) and GFP (Clones 1,2 and 3) were seeded in 6-well plate in three replica for each condition. After 24 h of seeding, cells were induced with doxycycline for 16 h. Cells were first washed with ice-cold PBS containing 1 mM $Na_3VO_4$ and 1 mM NaF and then lysed in a lysis buffer containing 8M Urea, 20 mM HEPES, 1 mM $Na_3VO_4$, 1 mM NaF, 1 mM B-Glycerol phosphate and 0.25 mM $Na_2H_2P_2O_7$. After incubation on ice for 5 min, the cells were then scraped and collected in Eppendorf tubes and stored at −80°C. For sample analysis, cell lysates were thawed, protein digested with trypsin, and phosphopeptides were enriched using $TiO_2$ as described in (Wilkes & Cutillas, 2017).

### Nanoflow-liquid chromatography tandem mass spectrometry (LC–MS/MS)

Dried samples were dissolved in 0.1% TFA (0.5 μg/μl) and run in a LTQ-Orbitrap XL mass spectrometer (Thermo Fisher Scientific) connected to a nanoflow ultra-high pressure liquid chromatography (UPLC, NanoAcquity, Waters). Peptides were separated using a 75 μm × 150 mm column (BEH130 C18, 1.7 μm Waters) using solvent A (0.1% FA in LC–MS grade water) and solvent B (0.1% FA in LC–MS grade ACN) as mobile phases. The UPLC settings consisted of a sample loading flow rate of 2 μl/min for 8 min followed by a gradient elution with starting with 5% of solvent B and ramping up to 35% over 100 min followed by a 10-min wash at 85% B and a 15-min equilibration step at 1% B. The flow rate for the sample run was 300 nL/min with an operating back pressure of about 3,800 psi. Full scan survey spectra ($m/z$ 375–1,800) were acquired in the Orbitrap with a resolution of 30,000 at $m/z$ 400. A data-dependent analysis (DDA) was employed in which the five most abundant multiply charged ions present in the survey spectrum were automatically mass-selected, fragmented by collision-induced dissociation (normalised collision energy 35%) and analysed in the LTQ. Dynamic exclusion was enabled with the exclusion list restricted to 500 entries, exclusion duration of 30 s and mass window of 10 ppm.

### Database search for peptide/protein identification and MS data analysis

Peptide identification was by searchers against the Swiss-Prot database (version 2013-2014) restricted to human entries using the

Mascot search engine (v 2.5.0, Matrix Science). The parameters included trypsin as digestion enzyme with up to two missed cleavages permitted, carbamidomethyl (C) as a fixed modification and Pyro-glu (N-term), Oxidation (M) and Phospho (STY) as variable modifications. Datasets were searched with a mass tolerance of $\pm$ 5 ppm and a fragment mass tolerance of $\pm$ 0.8 Da.

The automated programme Pescal (Cutillas & Vanhaesebroeck, 2007) was used to calculate the peak areas of the peptides identified by the mascot search engine. Proteins were identified with at least two peptides matched to the protein and a mascot score cut-off of 50 was used to filter false-positive detection. The resulting quantitative data were parsed into Excel files for normalisation and statistical analysis. Significance was assessed by $t$-test of $\log_2$ transformed data. When required, $P$-values were adjusted using the Benjamini–Hochberg method. Results are shown as $\log_2$ fold IKKε over control.

## Western blot

Protein levels were assessed using Western blotting. Cells were lysed in a lysis buffer (20 mM Tris–HCl, pH 7.4, 135 mM NaCl, 1.5 mM $MgCl_2$, 1% Triton, 10% glycerol) containing cOmplete protease inhibitor cocktail (Roche) and, where necessary, HALT phosphatase inhibitor cocktail (78428, Thermo Fisher Scientific). Samples were quantified using DC protein assay kit (Bio-Rad), and equal concentration samples were then prepared for SDS-PAGE in loading buffer (40% Glycerol, 30% β-Mercaptoethanol, 6% SDS, bromophenol blue). SDS-PAGE was performed using either 10 or 4–12% NuPAGE™ Bis-Tris Protein gels (Invitrogen) and resolved protein was transferred to Immobilon-P PVDF 0.45 μm Membrane (Merck). For immunoblotting, membranes were blocked for 1 h at room temperature in 5% w/v skimmed milk powder (Sigma-Aldrich) diluted in TBS-T solution (1× tris-buffered saline (TBS) (Severn Biotech) containing 0.1% v/v TWEEN® 20 (P1379, Sigma-Aldrich)) and then incubated overnight with primary antibodies diluted in 5% w/v milk in TBS-T at 4°C with constant agitation. Membranes were washed a minimum of three times over 15 min in TBS-T at room temperature before incubation with secondary antibodies diluted in 5% w/v milk in TBS-T at room temperature with constant agitation. Membranes were washed again prior to development with Pierce™ Enhanced Chemiluminescence Western Blotting Substrate (32106, Thermo Fisher Scientific), SuperSignal™ West Pico PLUS Chemiluminescence Substrate (34577, Thermo Fisher Scientific) or SuperSignal™ West Femto Maximum Sensitivity Substrate (34094, Thermo Fisher Scientific). Chemiluminescent signal was detected using either Fuji Medical X-Ray Film (Fujifilm) or an Amersham Imager 600UV chemiDoc system (GE Healthcare).

Primary antibodies used were as follows: Actin (sc-1615, Santa Cruz Biotechnology); ATF4 (ab1371, Abcam); c-Myc (Y69 clone, Abcam); HA-tag (11867423001, Roche); IKKε (14907, Sigma-Aldrich); IRF3 (11904, Cell Signaling); p-IRF3 Ser396 (4947, Cell Signaling); OAS1 (sc-98424, Santa Cruz Biotechnology); PHGDH (HPA021241, Sigma); PSAT1 (20180-1-AP, Proteintech Europe); PSPH (14513-1-AP, Proteintech Europe); SHMT2 (12762, Cell Signaling); P65 (8242, Cell Signaling); p-P65 (3039, Cell Signaling); STAT1 (9172, Cell Signaling); p-STAT1 Tyr701 (9167, Cell Signaling); Vinculin (66305-1-Ig, Proteintech Europe). Secondary antibodies used were as follows: Anti-mouse IgG, HRP-linked Antibody (7076, Cell Signaling); chicken anti-rat IgG-HRP (sc-2956, Santa Cruz

Biotechnology); donkey anti-goat IgG-HRP (sc-2020, Santa Cruz Biotechnology); goat anti-mouse IgG-HRP (sc-2005, Santa Cruz Biotechnology); goat anti-rabbit IgG-HRP (sc-2004, Santa Cruz Biotechnology); Rabbit IgG-HRP Linked Whole Ab (NA934, GE Healthcare).

All Western blots were performed with a minimum of three independent biological replicates, unless otherwise indicated in specific figure legends.

For densitometry analysis of Western blots in Fig 5C–E and Fig EV2D, relative protein band density was quantified using NIH's ImageJ software (Schneider et al, 2012). Vinculin protein band density was initially calculated for each sample. Then, within each cell line, the percentage of total density for control siRNA and IKKε siRNA transfected samples was calculated. This process was repeated to calculate relative densities for each protein of interest. Finally, the protein of interest percentage density was divided by the corresponding vinculin percentage density for each sample to generate normalised relative density values.

## High-content imaging and measurement of mitochondrial membrane potential ($\Delta\psi_m$)

Cells were seeded in thin, clear bottom black 96-well plates (BD Falcon) at medium density (4,000 cells/well) 24 h before the experiments. Prior to imaging cells were loaded with 1 μg/ml Hoechst 33342 (Sigma-Aldrich) and 30 nM tetramethyl-rhodamine-methylester (TMRM) for 30 min. TMRM was present during imaging in the solution (DMEM w/o phenol red). Images were acquired with the ImageXpress Micro XL (Molecular Devices) high-content wide field digital imaging system using a Lumencor SOLA light engine illumination, ex377/50 nm em447/60 nm (Hoechst) or ex562/40 nm and ex624/40 nm (TMRM) filters, and a 60X, S PlanFluor ELWD 0.70 NA air objective, using laser-based autofocusing. Sixteen fields/well were acquired. Images were analysed with the granularity analysis module in the MetaXpress 6.2 software (Molecular Devices) to find mitochondrial (TMRM) and nuclear (Hoechst) objects with local thresholding. Average TMRM intensities per cell were measured and averaged for each well. The mean of wells was then used as individual data for statistical analysis to compare each condition.

## PDH activity measurement

PDH activity was measured on whole cell lysates using the pyruvate dehydrogenase (PDH) Enzyme Activity Microplate Assay Kit (ab109902, Abcam).

## qRT–PCR

mRNA levels were assessed using quantitative real-time PCR (qRT–PCR). Total RNA was extracted from cells using the RNeasy Mini Kit (Qiagen) as per the manufacturer's protocol. RNA yield was quantified using the NanoDrop ND-1000 (Thermo Fisher Scientific), and 1 mg of RNA was reverse transcribed to cDNA using the Omniscript RT Kit (Qiagen). qPCR was performed using the TaqMan™ assay system.

The following TaqMan™ gene expression probes were used: *PHGDH* (Hs00198333_m1, Thermo Fisher Scientific); *PSAT1* (Hs00795278_mH, Thermo Fisher Scientific); *PSPH* (Hs00

190154_m1, Thermo Fisher Scientific); *ACTB* (β-Actin, 4310881E, Applied Biosystems).

Assay mixtures were prepared consisting of 10 μl TaqMan™ Master Mix (Applied Biosystems), 1 μl TaqMan™ gene probe & 1 μl cDNA, topped up to 20 μl with 8 μl RNase free $H_2O$. The qPCR reaction was carried out using either the 7500 Real Time or the QuantStudio 5 Real-Time PCR systems (Applied Biosystems), and the process was 2 min at 50°C, followed by 10 min holding at 95°C, then 40 cycles of 15 seconds at 95°C and 1 min at 60°C. Relative mRNA quantifications were obtained using the comparative Ct method, and data were analysed using either the 7500 software v2.3 or QuantStudio Design & Analysis Software (Applied Biosystems).

### Generation of conditioned medium

Flp-In 293 HA-GFP or HA-IKKε cells were treated for 16 h with 50 ng/ml doxycycline in 1 ml of medium per well of a 6-well plate, allowing secretion of potential signalling factors into the medium. Following induction, medium was collected and filtered using a 0.22 μM pore size filter and stored at 4°C till use.

### Gene expression analysis of clinical samples

The METABRIC dataset (Curtis *et al*, 2012) was obtained from Synapse: https://www.synapse.org/#!Synapse:syn1688369 (METABRIC Data for Use in Independent Research). All analysis was carried out using Bioconductor R packages. Overexpression of all genes was determined by fitting a Gaussian distribution to the central subpopulation shifted to zero and then determining samples which had expressions greater than 1.96 times the standard deviation from zero.

## Data availability

Datasets generated as part of this study through labelled metabolite analysis and phosphoproteomic analysis are both provided in full as part of this manuscript as Datasets EV1 and EV2, respectively.

### Statistical analysis

Data are presented as mean ± either standard deviation (SD) or standard error of the mean (SEM) as indicated in the figure legends. Statistical analysis tests were performed using GraphPad Prism (version 8), and specific tests were performed as indicated in the figure legends. Statistical significance was assumed at $P < 0.05$ and is noted on figures using *$P < 0.05$, **$P < 0.01$, ***$P < 0.001$ and ****$P < 0.0001$ where appropriate.

**Expanded View** for this article is available online.

### Acknowledgements

We are grateful to Dr. Alice Shia and Prof. Peter Schmid (BCI – QMUL) for providing the breast cancer cell line panel and to Dr. Tencho Tenev and Prof. Pascal Meier (ICR, London) for useful advice and discussion, to Dr Gunnel Hallden (BCI – QMUL) for providing reagents, to Prof. Ivan Dickic (Institute of Biochemistry II – Goethe University, Germany) for providing plasmids containing IKKε mutants and to the Barts Cancer Institute FACS facility for their support. KaB is supported by the Barts London Charity (Grant Reference Number: 467/2053), and WJ is supported by the Medical Research Council (MRC) PhD programme. PC is funded by the BBSRC (BB/M006174/1) and Barts and The London Charity (297/2249). GS is funded by University College London COMPLeX/British Heart Foundation Fund (SP/08/004), the BBSRC (BB/L020874/1) and the Wellcome Trust (097815/Z/11/Z) in the UK, and the Italian Association for Cancer Research (AIRC, IG22221). CF and ASHC are funded by the Medical Research Council, core fund to the MRC Cancer Unit (MRC_MC_UU_12022/6).

### Author contributions

RX performed experiments to characterise the effect of IKKε on mitochondrial oxygen consumption rate and to test the sensitivity of breast cancer cell lines to different drugs. WJ performed the experiments to characterise the mechanism through which IKKε regulates cellular metabolism (qRT-PCR and WB). RX did the cloning of IKKε mutants and generated the cell lines. EW-V helped with the phosphoproteomic experiment together with VR and PC that also performed the MS for the *in vitro* kinase assay. ASHC and CF performed the MS experiment with metabolic tracers and analysed the data. AN and CC helped with phosphoproteomic data analysis. BY contributed to the OCR measurements, SOB to the characterisation of the role of ATF4 and YW to the cloning. GS performed the experiments to measure mitochondrial membrane potential and analysed the data. RBB, GS and KeB analysed the gene expression datasets. KaB designed the study and wrote the manuscript with the help of RX, WJ, PC, GS and CF.

### Conflict of interest

The authors declare that they have no conflict of interest.

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
