## [Review Process File · EMBO Reports]

The breast cancer oncogene IKK ϵ coordinates mitochondrial function and serine metabolism

Ruoyan Xu, William Jones, Ewa Wilcz-Villega, Sofia Costa, Vinothini Rajeeve, Robert Bentham, Kevin Bryson, Ai Nagano, Busra Yaman, Sheila Olendo Barasa, Yewei Wang, Claude Chelala, Pedro Cutillas, Gyorgy Szabadkai, Christian Frezza, and Katuscia Bianchi
DOI: 10.15252/embr.202050800

Corresponding author(s): Katuscia Bianchi (k.bianchi@qmul.ac.uk)

Review Timeline:

Submission Date:	10th Apr 19
Editorial Decision:	10th Apr 19
Revision Received:	6th Jul 19
Editorial Decision:	13th Aug 19
Revision Received:	14th Sep 19
Editorial Decision:	5th Nov 19
Revision Received:	4th May 20
Editorial Decision:	10th Jun 20
Revision Received:	29th Jun 20
Accepted:	9th Jul 20

Editor: Achim Breiling

Transaction Report: This manuscript was transferred to EMBO reports following peer review at The EMBO Journal.

Dear Dr. Bianchi,

Thank you for the transfer of your research manuscript to EMBO reports. I have now read your revised paper and went through the referee reports from The EMBO Journal (which you will find attached at the end of this message).

All referees acknowledge that the revised manuscript has improved, and that many concerns have been addressed. However, referees #2 and #3 state that part of their concerns have not been adequately addressed, and referee #2 also points out further concerns and suggestions to improve the manuscript.

Nevertheless, I would be happy to receive a final revised version of the manuscript. As EMBO reports emphasizes novel functional over detailed mechanistic insight, requests from the referee asking for further mechanistic data don't need to be addressed (if you already have such data, though, we ask you to add these to the revised manuscript). But remaining concerns regarding the major message of this study (including technical issues), in particular the remaining points referee #2, need to be addressed experimentally and/or in a detailed point-by-point response. In this case, it might also be advisable to shorten the paper, and to leave out data the referees (or you) feel do not support the conclusions, or are not central to the study.

Given the constructive referee comments, we would like to invite you to revise your manuscript for EMBO reports with the understanding that the referee concerns must be addressed in the revised manuscript (as indicated above) and in a detailed point-by-point response. Acceptance of your manuscript will depend on a positive outcome of a final round of review (involving referee #2). It is our policy to allow a single round of revision only and acceptance or rejection of the manuscript will therefore depend on the completeness of your responses included in the next, final version of the manuscript.

Revised manuscripts should be submitted within three months of a request for revision; they will otherwise be treated as new submissions. Please contact us if a 3-months time frame is not sufficient for the revisions so that we can discuss the revisions further.

Please refer to our guidelines for preparing your revised manuscript and the figure panels:

<http://embor.embopress.org/authorguide#manuscriptpreparation>

http://embopress.org/sites/default/files/EMBOPress_Figure_Guidelines_061115.pdf

Supplementary/additional data: The Expanded View format, which will be displayed in the main HTML of the paper in a collapsible format, has replaced the Supplementary information. You can submit up to 5 images as Expanded View. Please follow the nomenclature Figure EV1, Figure EV2 etc. The figure legend for these should be included in the main manuscript document file in a section called Expanded View Figure Legends after the main Figure Legends section. Additional Supplementary material should be supplied as a single pdf labeled Appendix. The Appendix includes a table of content on the first page, all figures and their legends. Please follow the nomenclature Appendix Figure Sx throughout the text and also label the figures according to this nomenclature. For more details please refer to our guide to authors.

Important: All materials and methods should be included in the main manuscript file.

Regarding data quantification and statistics, can you please specify, where applicable, the number "n" for how many independent experiments (biological replicates) were performed, the bars and error bars (e.g. SEM, SD) and the test used to calculate p-values in the respective figure legends. Please provide statistical testing where applicable. Please also provide a paragraph in the methods sections detailing the statistics used throughout the manuscript. See also: <http://embor.embopress.org/authorguide#statisticalanalysis>

We now strongly encourage the publication of original source data with the aim of making primary data more accessible and transparent to the reader. The source data will be published in a separate source data file online along with the accepted manuscript and will be linked to the relevant figure. If you would like to use this opportunity, please submit the source data (for example scans of entire gels or blots, data points of graphs in an excel sheet, additional images, etc.) of your key experiments together with the revised manuscript. Please include size markers for scans of entire gels, label the scans with figure and panel number. Please send one PDF file per figure.

Please add a short running title (up to 40 characters, including spaces) to the title page of the manuscript.

Finally, please format the references according to our journal style. See: <http://embor.embopress.org/authorguide#referencesformat>

- a complete author checklist, which you can download from our author guidelines (<http://embor.embopress.org/authorguide#revision>). Please insert page numbers in the checklist to indicate where the requested information can be found.
- a letter detailing your responses to the referee comments in Word format (.doc)
- a Microsoft Word file (.doc) of the revised manuscript text
- editable TIFF or EPS-formatted single figure files in high resolution (for main figures and EV figures)

I look forward to seeing a revised version of your manuscript when it is ready. Please let me know if you have questions or comments regarding the revision.

Kind regards,

Achim

Achim Breiling
Editor
EMBO reports

Referee #1:

The authors have responded to my previous critiques by providing extensive new data in their rebuttal letter. These data support the conclusions and answer the concerns. The response to the other reviewers' concerns is very detailed and the paper has been significantly improved.

Referee #2:

While the manuscript revisions and rebuttal authors address multiple issues in the original review, there are several issues that need to be resolved.

1) The issues relating to the mechanism whereby IKK ϵ regulates PSAT1 phosphorylation or expression have not been resolved.

Ref 2 point 2: The authors attempted to address whether TBK is required for the phosphorylation of PSAT, however they ran into technical difficulties and were unable to carry out in vitro kinase assays using TBK knock-out cells. Despite this, the authors state "in the revised version of the manuscript we make clear that it is the IKK ϵ /TBK1 complex that is responsible for phosphorylating PSAT1." This conclusion is not appropriate as there is no data supporting a role for TBK other than it is present in IPs of IKK ϵ as detected by mass spec.

In addition, the use of an in vitro kinase assaying using HA-tagged IKK ϵ does not prove that "IKK ϵ was required for PSAT1 phosphorylation (S331)". This result just shows that there is not a contaminating PSAT1 kinase in their in vitro kinase assay. Another confusing point relating to these findings is that the evidence that kinase dead IKK ϵ is able to phosphorylate PSAT1 in vitro implicates an associated kinase, but if this were the case, it is baffling that the site of phosphorylation on PSAT1 is a consensus IKK ϵ site.

The authors argue that their data in Figure 3 shows that phosphorylation of PSAT1 doesn't regulate its activity and thus is not relevant to the increased levels of serine biosynthesis. This is true; however, this finding makes this whole section of the paper on phosphorylation irrelevant to the overall findings and clearly doesn't justify the conclusions drawn the this paper provides the "first evidence that "the IKK ϵ /TBK1 complex is the first reported kinase to phosphorylate PSAT1."

2) Since basal/ER-negative tumors express high levels of PSAT1 and PSAT1 is methylated and repressed in most ER+ tumors, it is important to add an additional analysis to ensure that correlations between PSAT1 and mitochondrial gene expression (and IKK ϵ) are clearly independent of subtype by carrying out separate analyses of the basal tumors alone. It isn't clear that this was done in the analysis. It would

3) The correlation analysis with co-regulated mitochondrial genes would be more relevant is indeed the same set of mitochondrial genes were shown to be regulated in the tissue culture lines examined under conditions where IKK ϵ was genetically manipulated.

Referee #3:

In the revised manuscript, the authors have performed a series of experiments to strengthen the conclusions of the previous manuscript. The number of experiments and amount of data the authors present is impressive but unfortunately, they fail to present novel mechanisms that would significantly advance the field. Importantly, the authors propose that IKK ϵ regulates mitochondrial function which in turn induces serine biosynthesis. However, the authors don't present data that would link IKK ϵ to the regulation of mitochondrial function. The current data indicate changes in mitochondrial oxygen consumption rate, but mechanistically this is not detailed enough to advance

the existing literature. The authors identify ATF4 as the regulator of the process which together with upregulation of serine synthesis indeed indicates the involvement of mitochondrial metabolism. However, this link has already been robustly shown in the literature, so a significant contribution to the field would be to address the question, how does IKK regulate mitochondrial function?

In addition, what is the mechanism the authors propose that leads to decreased serine synthesis in cancer cell lines where IKK ϵ was knocked down? Previous reports have shown that ATF4 mediates a transcriptional program that is activated in mtDNA mutant models but currently it remains unclear how this pathway could be involved in IKK ϵ knock-down cell model system.

Re: EMBOR-2019-48260-T

Referee#2:

While the manuscript revisions and rebuttal authors address multiple issues in the original review, there are several issues that need to be resolved.

1) The issues relating to the mechanism whereby IKK ϵ regulates PSAT1 phosphorylation or expression have not been resolved. Ref 2 point 2: the authors attempted to address whether TBK is required for the phosphorylation of PSAT, however they ran into technical difficulties and were unable to carry out in vitro kinase assays using TBK knock-out cells. Despite this, the authors state "in the revised version of the manuscript we make clear that it is the IKK ϵ /TBK1 complex that is responsible for phosphorylating PSAT1." This conclusion is not appropriate as there is no data supporting a role for TBK other than it is present in IPs of IKK ϵ as detected by mass spec.

In addition, the use of an in vitro kinase assaying using HA-tagged IKK ϵ does not prove that "IKK ϵ was required for PSAT1 phosphorylation (S331)". This result just shows that there is not a contaminating PSAT1 kinase in their in vitro kinase assay.

We agree with the referee that no conclusions on the role of TBK1 on PSAT phosphorylation can be drawn from these experiments, and have removed the conclusion. We have changed the text to "a kinase complex, containing IKK ϵ and TBK1, was capable of phosphorylating PSAT1".

Another confusing point relating to these findings is that the evidence that kinase dead IKK ϵ is able to phosphorylate PSAT1 in vitro implicates an associated kinase, but if this were the case, it is baffling that the site of phosphorylation on PSAT1 is a consensus IKK ϵ site.

To avoid confusion on this point we have underlined that the identified consensus sequence for TBK1 is identical to the one of IKK ϵ (Hutti *et al*, 2012).

The authors argue that their data in Figure 3 shows that phosphorylation of PSAT1 doesn't regulate its activity and thus is not relevant to the increased levels of serine biosynthesis. This is true; however, this finding makes this whole section of the paper on phosphorylation irrelevant to the overall findings and clearly doesn't justify the conclusions drawn in this paper. The paper provides the 'first evidence that "the IKK ϵ /TBK1 complex is the first reported kinase to phosphorylate PSAT1.

We recognise that PSAT1 phosphorylation is not the mechanism by which IKK ϵ regulates the SBP and have accordingly toned down results related to the phosphorylation. However, we also think that it is important to report that we have indeed shown that IKK ϵ can phosphorylate PSAT1. Although this phosphorylation is not required for its enzymatic function, it may be implicated in other functions of PSAT1, which we are currently investigating.

2) Since basal/ER-negative tumors express high levels of PSAT1 and PSAT1 is methylated and repressed in most ER+ tumors, it is important to add an additional analysis to ensure that correlations between PSAT1 and mitochondrial gene expression (and IKK ϵ) are clearly independent of subtype by carrying out separate analyses of the basal tumors alone. It isn't clear that this was done in the analysis.

As we discussed in the previous rebuttal letter (referee#2, point 7), we agree with the reviewer that a correlation between the ER status of the tumours and their IKK ϵ , PSAT1 and mitochondrial gene expression exists. However, we also explained that we identified a subset of basal ER- negative tumours where SBP genes are induced, and this correlates with the suppression of a mitochondrial gene set (gene group 1). We have further confirmed these findings by showing that

1. the overall correlation between IKK ϵ and PSAT1 (as shown in Fig. EV7 A,B and Fig. 1A below) applies also to ER- negative tumours ($p=0.02$), and
2. Among the ER- negative tumours, IKK ϵ and PSAT1 expression is significantly higher in the samples where mitochondrial gene group 1 is suppressed (i.e. samples in the upper fork of the sample distribution) (Fig. 1B below).

Fig. 1: (A) correlation between the expression of IKK ϵ (*IKBKE*) and PSAT1 in ER- samples; **(B)** comparison of the level of expression of IKK ϵ in ER- samples in the upper fork (UF) or outside the fork.

These results confirm that the regulation of mitochondrial genes and PSAT1 by IKK ϵ is partly independent of subtypes. However, we do not exclude that this regulation also correlates with ER positivity. The exact causal relationship between these factors needs further experimental validation.

3) The correlation analysis with co-regulated mitochondrial genes would be more relevant if indeed the same set of mitochondrial genes were shown to be regulated in the tissue culture lines examined under conditions where IKK ϵ was genetically manipulated.

Since we had no opportunity to perform transcriptomic analysis in the cell lines where IKK ϵ was silenced, we used a publicly available microarray dataset from a collection of 51 breast cancer cell lines (Neve, R. M., Chin, K., Fridlyand, J., Yeh, J., Baehner, F. L., Fevr, T., Clark, L., Bayani, N., Coppe, J.-P., Tong, F. et al. (2006), 'A collection of breast cancer cell lines for the study of functionally distinct cancer subtypes', *Cancer cell* 10(6), 515–527.). In order to address the reviewer's question we have quantified the average expression of genes in both mitochondrial gene sets (group1 and group2, see Fig. EV7C), genes of the SBP (PSAT1, PHGDH and PSPH) and IKBKE. We noted that IKBKE mRNA does not show strong correlation with protein levels measured by western blot (please see Fig. 2 below), thus we explored correlation between the three parameters (mitochondrial gene sets, SBP and IKK ϵ) in two steps. First we have shown that the average expression of mitochondrial group1 significantly correlates with PSAT1 and PHGDH (Fig. EV7I) across all 51 breast cancer cell lines. Then, based on our western blot data, we have selected cell lines with high (MDA-MB-231, HCC143, MDA-MB-468, MDA-MB-175) and low (HCC38, SKBR3, SUM159PT, HCC1954, MDA-MB-453, ZF-75.1) IKK ϵ expression, and compared the expression of SBP enzymes in the two groups (Fig. EV7J). The results indicated that the same pathways operate in cellular models.

Fig. 2: WB showing the level of expression of IKK ϵ in a panel of breast cancer cell lines.

Dear Katuscia,

Thank you for the submission of your revised manuscript to EMBO reports. We have now received the comments from the referee that was asked to re-assess it, which you will find below.

I am sorry to say that the evaluation of your revised manuscript is not a positive one. However I have taken the liberty to discuss your study with my colleague Andrea Leibfried at our sister journal LSA, and she would be happy to offer publication of this work essentially without the addition of further experimental data (taking into account of the previous rounds of review at The EMBO Journal and EMBO reports), if you were to transfer your manuscript there (see the transfer link below). However, Andrea indicated that the manuscript needs further re-shaping and shortening, including the removal of the PSAT1 data and toning down the conclusions based on the tumor correlation analysis. Also the remaining concerns of the referee should be addressed in a point-by-point response and by changing the manuscript accordingly.

Regarding EMBO reports, however, as you will see, the referee states that the issues raised in her/his last report were not addressed in a constructive way. The referee still questions that part of the data are biologically meaningful, and doubts a direct link between IKKe and PSAT1.

Given these remaining concerns, the fact that you already had a chance to significantly revise the study, and that we allow a single round of revision only, I am afraid that we cannot offer to publish the manuscript at this point. I am sorry that this final decision emerges as the outcome of a lengthy review process, but given that the referee is still not convinced by the current set of data, I have no other option but to reject your manuscript.

That being said, as mentioned above, Life Science Alliance would like to continue with this manuscript. Life Science Alliance is launched as a partnership between EMBO Press, Rockefeller Press, and Cold Spring Harbor Laboratory Press, and publishes work that is of high value to the respective communities across all areas in the life sciences (see: <http://www.life-science-alliance.org>). As indicated, Andrea would be happy to offer publication of this work after further revisions.

If you are interested in this option - please use the link below for transfer.

If you would like to inquire with Andrea first, please contact her at: a.leibfried@life-science-alliance.org

Kind regards,

Achim

Achim Breiling
Editor
EMBO Reports

Referee #1 (Referee #2 from the submission at The EMBO Journal):

The authors have not addressed the issues raised in the last review in a constructive way.

Point 1 from previous review:

Data on phosphorylation of PSAT by IKKe. While we can understand that the authors want to publish the in vitro phosphorylation data, in the absence of other data supporting the in vivo phosphorylation of PSAT by IKKe and the relevance of this phosphorylation on function, this finding is very confusing in the context of this manuscript.

Point 2 from previous review:

Correlation between IKKe and PSAT1 in ER- tumors: The correlations shown for IKKe and PSAT1 in ER- tumors in the rebuttal and manuscript are extremely weak ($r^2 = 0.01$) and highly questionable as being meaningful biologically, even if the p value is 0.02 (which is mostly a reflection of the very large number of samples analyzed). Several other r^2 values in the manuscript are $<.05$ and similarly of questionable relevance.

Point 3 from previous review:

"The correlation analysis with co-regulated mitochondrial genes would be more relevant if indeed the same set of mitochondrial genes were shown to be regulated in the tissue culture lines examined under conditions where IKKe was genetically manipulated."

The correlation analysis of pre-existing published data from cell lines (which is provided in the manuscript) isn't a good substitute for analysis of genes that are regulated by IKKe using genetic manipulations similar to those in this report and doesn't address this point effectively. More importantly, the rationale for investigating expression of mitochondrial genes isn't clear. The authors show that the effect of IKKe on mitochondria is through phosphorylation of PDHA1. It is therefore unclear why they shift to an analysis of mitochondrial gene expression, especially since there isn't any evidence that mitochondrial gene expression is regulated by IKKe. Because of this, it isn't clear how the mitochondrial gene correlation data relates to the IKKe/PDH mechanism that is central to the study. This also applies to the complex computational analysis shown in Figure 7. The authors state that "we took into account that the pathway linking IKKe to PSAT1 transcriptional regulation was mitochondrial-nuclear retrograde signaling." What evidence is there that this is mitochondrial-nuclear retrograde signaling? The preceding data suggests that IKKe directly phosphorylates and inhibits PDH; therefore, one question is if there is a need to invoke a complex pathway involving IKKe regulation of PSAT1, which in turn regulates mitochondrial gene expression? It seems far more likely that IKKe directly regulates PDH activity and that the PSAT1/mitochondrial gene connections are unrelated.

Other points:

1) Figure 1:

- a) Is there a reason that the authors only show citrate m+4? There are 6 carbons in citrate. Does the absence of this data indicate that there was 0% label in m+5 and m+6?
- b) There are significant levels of m+1 and m+2 serine. Do the authors understand the source of this? It isn't clear where this would come from. Also, the authors conclude that there is an increase in uptake of unlabeled serine from the media. What is the basis for this conclusion. As presented, the metabolites that appear to be changing are the unexplained m+1 and m+2 serine fractions.
- c) The authors indicate that the decrease in m+2 citrate and malate is indicative of a decrease in TCA cycle anaplerosis. Decreased m+2 is indicative of decreased PDH activity, which is not anaplerosis. Anaplerosis from glucose derived carbon would most likely be indicated by m+3 as an

indicator of pyruvate carboxylase activity.

d) Also worth noting is that in Figure 1C, a doubling of m+3 serine is reported, but there doesn't seem to be any difference in m+3 in Figure 1F under what looks like the same conditions.

2) At the end of page 6, the authors state that "the fraction of $^{15}\text{N}_2$ labelled serine, derived from glutamine was also increased, indicating higher glutamine usage in serine biosynthesis as a nitrogen source". Was this meant to be "the fraction of ^{15}N -labeled serine" since serine has only one nitrogen. In addition, what is the difference between the ^{15}N and m+1 label on Figure 1G? One ^{15}N should make m+1 serine. Perhaps the LCMS is capable of differentiating a +1 from ^{15}N vs ^{13}C , but in that case the simpler "m+1, m+2, etc" labeling should not be used.

3) Is there a reason that the data shown in Figures 2E and 2F is represented as "ion intensity"? This is different than Figure 1, where they show it as fractional labeling? This needs clarification. Also, in figure 2E the authors show glycine, citrate, and malate, and in figure 2F they show serine, glycine, serine and malate. Why were these metabolites chosen? Why did they show some metabolites from one cell line but not from the other?

4) After describing the data in figure 2, the authors state: "Taken together, these data indicate that in cancer cells IKKe redirects a significant fraction of glucose-derived carbons to the SBP, at the expense of pyruvate oxidation in the TCA cycle." This conclusion is not well supported by the data and draws inferences that aren't appropriate without quantitative flux experiments (or at least experiments which show that serine synthesis pathway inhibition normalizes the changes in pyruvate metabolism and mitochondrial function). This is a particularly puzzling conclusion because the authors subsequently present evidence that IKKe directly controls PDH activity, which would be sufficient to regulate pyruvate levels without invoking the serine synthesis pathway.

5) The authors cite the Possemato et al report stating that knockdown of PHGDH "abolishes proliferation of cells even in complete medium". This report showed that cells respond differently to PHGDH knockdown - some are inhibited a great deal, others to a lesser degree. Thus, the inference about how the 293T cells used in their report "should" be functioning based on this paper is not appropriate.

** As a service to authors, EMBO Press provides authors with the ability to transfer a manuscript that one journal cannot offer to publish to another journal, without the author having to upload the manuscript data again. To transfer your manuscript to another EMBO Press journal using this service, please click on Link Not Available

Re: EMBOR-2019-48260V2 - Rebuttal Letter

From email – 13th August 2019

Point 1 from previous review:

Data on phosphorylation of PSAT by IKKe. While we can understand that the authors want to publish the in vitro phosphorylation data, in the absence of other data supporting the in vivo phosphorylation of PSAT by IKKe and the relevance of this phosphorylation on function, this finding is very confusing in the context of this manuscript.

We have tried to change the narrative of the paper in multiple ways, however we believe the current version is the most clear. Importantly, the other two referees did not find this part confusing. However, if you - as editor - also think the paper would benefit from changing this part we will follow your advice.

Point 2 from previous review:

Correlation between IKKe and PSAT1 in ER- tumors: The correlations shown for IKKe and PSAT1 in ER- tumors in the rebuttal and manuscript are extremely weak ($r^2 = 0.01$) and highly questionable as being meaningful biologically, even if the p value is 0.02 (which is mostly a reflection of the very large number of samples analyzed). Several other r^2 values in the manuscript are $<.05$ and similarly of questionable relevance.

Analysis of the ER- set of the METABRIC dataset (Fig. 7G) gave a correlation value of $R^2=0.39$ ($p<0.001$) between IKBKE and PSAT1 expression in 50 samples. Thus, it is unclear what the reviewer is referring to. As supplementary analysis, Fig. EV7A,B shows the results of two independent methods to analyse the co-expression of the two proteins in the whole METABRIC dataset including both ER+ and ER- tumours. Here, whilst a Pearson's test (panel A) indeed indicated low correlation ($R^2=0.04$, $p<0.001$), chi-square independence test (panel B) gave a robust indication that IKBKE and PSAT1 are co-expressed in a statistically highly significant and pathologically relevant fraction (6.1%) of tumours. The only other low correlation values in the paper are the correlation of PHGDH and PSPH, which are shown for information, and even if significant, no conclusions were drawn from those data.

In addition – please also see our answer below to the email of the 23rd of August.

Point 3 from previous review:

"The correlation analysis with co-regulated mitochondrial genes would be more relevant if indeed the same set of mitochondrial genes were shown to be regulated in the tissue culture lines examined under conditions where IKKe was genetically manipulated."

The correlation analysis of pre-existing published data from cell lines (which is provided in the manuscript) isn't a good substitute for analysis of genes that are regulated by IKKe using genetic manipulations similar to those in this report and doesn't address this point effectively.

Within the limited timeline of the revision, we could not perform RNA-seq experiments. Therefore, we relied on publicly available datasets. This approach also offered validation of our findings in an independent dataset. It is unclear why the referee finds this well-established orthogonal validation not suitable.

More importantly, the rationale for investigating expression of mitochondrial genes isn't clear. The authors show that the effect of IKKe on mitochondria is through phosphorylation of PDHA1. It is therefore unclear why they shift to an analysis of mitochondrial gene expression, especially since there isn't any evidence that mitochondrial gene expression is regulated by IKKe. Because of this, it isn't clear how the mitochondrial gene correlation data relates to the IKKe/PDH mechanism that is central to the study. This also applies to the complex computational analysis shown in Figure 7.

Our data analysis clearly shows that there is strong negative correlation between IKBKE expression and the expression of a specific mitochondrial gene group (group1) both in patient samples (Fig. EV7G, $R^2=0.49$, $p<0.001$) and cell lines (Fig. EV7I, $R^2=0.47$, $p<0.001$). The main rationale to include the investigation of mitochondrial genes was that the strongest correlation between IKBKE and PSAT1 can be found in the samples where mitochondrial genes are also regulated (Fig. 7G). Although this result will require further investigation, it corroborated the relationship between IKKe and serine metabolism.

The authors state that "we took into account that the pathway linking IKKe to PSAT1 transcriptional regulation was mitochondrial-nuclear retrograde signaling." What evidence is there that this is mitochondrial-nuclear retrograde signaling?

The evidence for retrograde signalling is twofold: first, ATF4 is required for PSAT1 upregulation (Fig. 6B-E). Second, IKKe inhibits mitochondrial function (Fig. 5) and upregulates ATF4. In addition, there is strong evidence from previous studies (Picard *et al*, 2014; Bao *et al*, 2016; Khan *et al*, 2017) showing ATF4 as the target of retrograde signalling and key regulator of the SBP.

The preceding data suggests that IKKe directly phosphorylates and inhibits PDH; therefore, one questions if there is a need to invoke a complex pathway involving IKKe regulation of PSAT1, which in turn regulates mitochondrial gene expression? It seems far more likely that IKKe directly regulates PDH activity and that the PSAT1/mitochondrial gene connections are unrelated.

We disagree with this last point. The preceding data do NOT indicate that IKKe directly phosphorylates PDH for two main reason: first, the identified p-site does not match the one reported for IKKe, second PDH is in the mitochondrial matrix, while IKKe is mainly a cytosolic protein reported to be associate with the mitochondrial outer membrane. The "complex" pathway we "invoke" is not the one the referee refers to. Indeed, we showed that the modulation of PSAT1 does not affect mitochondrial metabolism (see Fig. 5D-E). We demonstrate that IKKe regulates the mitochondria and in turn the SBP enzymes, including PSAT1. We believe that it has been clearly explained in the manuscript in multiple points and find rather worrying that at this stage of the revision this referee yet does not grasp the main message of the paper.

Other points:

1) Figure 1:

a) Is there a reason that the authors only show citrate m+4? There are 6 carbons in citrate. Does the absence of this data indicate that there was 0% label in m+5 and m+6?

These metabolites were not detected in our analysis.

b) There are significant levels of m+1 and m+2 serine. Do the authors understand the source of this? It isn't clear where this would come from. Also, the authors conclude that there is an increase in uptake of unlabeled serine from the media. What is the basis for this conclusion. As presented, the metabolites that appear to be changing are the unexplained m+1 and m+2 serine fractions.

The source of m+1 and m+2 serine is unclear, even though it might be the results of scrambling of carbons between serine synthesis and catabolism. However, these isotopologues are not the focus of our work and we do not wish to speculate upon them. The conclusion that the uptake of serine is increased is based on the increased % of m+0 serine (unlabelled, coming from the medium) in the total pool.

c) The authors indicate that the decrease in m+2 citrate and malate is indicative of a decrease in TCA cycle anaplerosis. Decreased m+2 is indicative of decreased PDH activity, which is not anaplerosis. Anaplerosis from glucose derived carbon would most likely be indicated by m+3 as an indicator of pyruvate carboxylase activity.

The referee is correct and we amended the text accordingly.

d) Also worth noting is that in Figure 1C, a doubling of m+3 serine is reported, but there doesn't seem to be any difference in m+3 in Figure 1F under what looks like the same conditions.

The data shown in Fig. 1C represent the measurement of serine m+3, while the data shown in Fig. 1F represent the fractional enrichment of the total serine pool. Please note that this referee previously requested to add the fractional enrichment of serine.

2) At the end of page 6, the authors state that "the fraction of ¹⁵N₂ labelled serine, derived from glutamine was also increased, indicating higher glutamine usage in serine biosynthesis as a nitrogen source". Was this meant to be "the fraction of ¹⁵N-labeled serine" since serine has only one nitrogen.

The referee is correct, this was a typo and has been corrected.

In addition, what is the difference between the ¹⁵N and m+1 label on Figure 1G? One ¹⁵N should make m+1 serine. Perhaps the LCMS is capable of differentiating a +1 from ¹⁵N vs ¹³C, but in that case the simpler "m+1, m+2, etc" labeling should not be used.

Our LC/MS platform enables the distinction between ¹³C and ¹⁵N. We modified the figure to indicate C¹³ and N¹⁵ fractions.

3) Is there a reason that the data shown in Figures 2E and 2F is represented as "ion intensity"? This is different than Figure 1, where they show it as fractional labeling? This needs clarification. Also, in figure 2E the authors show glycine, citrate, and malate, and in figure 2F they show serine, glycine, serine and malate. Why were these metabolites chosen? Why did they show some metabolites from one cell line but not from the other?

The data are shown as "ion intensity" also in Fig1, please see Fig. 1C and E. The fractional enrichment shown in Fig. 1F was added as requested previously by this referee. In Fig. 2C and D all metabolites measured in T47D and MDA-MB-468 cells are shown, so we did not only show a few selected metabolites. Moreover, all data are also reported in Table 1.

4) After describing the data in figure 2, the authors state: "Taken together, these data indicate that in cancer cells IKKe redirects a significant fraction of glucose-derived carbons to the SBP, at the expense of pyruvate oxidation in the TCA cycle." This conclusion is not well supported by the data and draws inferences that aren't appropriate without quantitative flux experiments (or at least experiments which show that serine synthesis pathway inhibition normalizes the changes in pyruvate metabolism and mitochondrial function). This is a particularly puzzling conclusion because

the authors subsequently present evidence that IKKe directly controls PDH activity, which would be sufficient to regulate pyruvate levels without invoking the serine synthesis pathway.

We agree with the referee that this sentence might have been misleading. We modified the text as follows: Taken together, these data indicate that in cancer cells IKKe redirects a significant fraction of glucose-derived carbons to the SBP and reduces pyruvate oxidation in the TCA cycle."

5) The authors cite the Possemato et al report stating that knockdown of PHGDH "abolishes proliferation of cells even in complete medium". This report showed that cells respond differently to PHGDH knockdown - some are inhibited a great deal, others to a lesser degree. Thus, the inference about how the 293T cells used in their report "should" be functioning based on this paper is not appropriate.

Thank you, we have toned down this sentence.

From email – 23rd August 2019

"The weak correlation between IKBKE and PSAT that I was referring to in the review was based on the figure shown in the original rebuttal letter (rebuttal Figure 1(A) – rebuttal letter July 6th). They indicate that the left panel shows the correlation of IKKE and PSAT in ER negative tumors, and it clearly shows a very weak correlation ($r^2 = 0.012$). This is the basis for the point made in my review."

The $R^2 = 0.012$ value in the ER- subpopulation is the coefficient of variation, which indicates a correlation of $R = 0.11$. This is a weak but significant correlation. We have performed further linear modelling of the expression of IKBKE and PSAT1 in both the whole METABRIC dataset and in the ER-subset of samples and it confirmed that IKBKE expression is a strongly significant predictor of PSAT1 expression in both cases. See results below:

On the whole dataset:

```
Call:
lm(formula = PSAT1 ~ IKBKE, data = data)

Residuals:
    Min     1Q   Median     3Q      Max
-1.0474 -0.3802 -0.1893  0.2213  2.4367

Coefficients:
            Estimate Std. Error t value Pr(>|t|)
(Intercept)  0.21255    0.01289   16.495 <2e-16 ***
IKBKE        0.28716    0.02939    9.771 <2e-16 ***
---
Signif. codes:  0 '***' 0.001 '**' 0.01 '*' 0.05 '.' 0.1 ' ' 1

Residual standard error: 0.5718 on 1979 degrees of freedom
Multiple R-squared:  0.04602, Adjusted R-squared:  0.04554
F-statistic: 95.47 on 1 and 1979 DF, p-value: < 2.2e-16
```

On the ER- dataset:

```
Call:
lm(formula = PSAT1 ~ IKBKE, data = data2)

Residuals:
    Min     1Q   Median     3Q      Max
-1.33858 -0.38755  0.01209  0.40806  1.70016

Coefficients:
            Estimate Std. Error t value Pr(>|t|)
(Intercept)  0.92438    0.03245   28.484 <2e-16 ***
IKBKE        0.13716    0.05890    2.329  0.0203 *
---
Signif. codes:  0 '***' 0.001 '**' 0.01 '*' 0.05 '.' 0.1 ' ' 1

Residual standard error: 0.626 on 433 degrees of freedom
Multiple R-squared:  0.01237, Adjusted R-squared:  0.01009
F-statistic: 5.423 on 1 and 433 DF, p-value: 0.02034
```

In addition, we have performed multiple regression analysis, including known genes regulating PSAT1, such as ATF4 and MYC (code and data can be provided if requested). These genes showed

similar significance levels and the combined coefficient of variation was 0.14. Moreover, when we include IKBKE in the multiple regression model, it leads to significant increase in the coefficient of variation, indicating the IKBKE is an important predictor of PSAT1 expression.

The relatively low correlation between IKBKE and PSAT1 is due to the multiple regulation of PSAT1. PSAT1 is upregulated in substantial number of tumours, but only a subset of these is due to IKBKE.

On the implications of the claim by this reviewer that stronger correlation should be necessary to show in the ER- subgroup see our detailed comment below.

In the current rebuttal, they refer to Figure 7G as showing the data on the correlation between IKBKE and PSAT. But, I don't think that Figure 7G addresses the question. The correlation value for that figure is for the 'first 50 of ranking' - this isn't appropriate - the whole set of ER-negative tumors should be analyzed as they did in the rebuttal figure. They don't show the correlation for the basal tumors which are highlighted. Also for the basal tumors in EV7G (shown in red), there is no correlation between IKBKe and mitochondrial genes.

Figure EV7A and B includes data from all breast tumors, not ER negative/basal tumors. This analysis isn't meaningful on its own because the question is whether there is a correlation within ER-negative tumors. The rebuttal figure where ER-samples alone are analyzed (Fig. 1(A)) critically addresses whether PSAT and IKBKE correlations are independent of tumor subtype and this showed a non-meaningful extremely low correlation (r^2 0.01). There are many reports that show how the expression of genes in ER+ and ER- tumors are dramatically different due to the different lineages of these two different tumor types. So it is critical to look within each tumor type to look for meaningful correlations and the figure in the rebuttal did just that.

We assume that the reviewer requires correlation between PSAT1 and IKBKE in the basal, ER- subset of tumours is based on misunderstanding of our claims. Our aim with the bioinformatic analysis of IKBKE and PSAT1 expression was to simply show that there is a subgroup of tumours, where IKBKE and PSAT1 are co-expressed. While this has already been shown to be true for ER- tumours, our additional analysis using biclustering of mitochondrial gene expression shows that the co-expression of these genes is the strongest in a subgroup of tumours (Fig. 7G – 'first 50 of the ranking'), where mitochondrial gene expression is regulated. We obviously can not imply causality (IKBKE -> mitochondrial genes -> PSAT1) from gene expression data, thus we only claim that these data are consistent with the mechanistic pathway we identified by functional experiments.

We believe that the reviewer would require strong, subtype dependent correlation between IKBKE and PSAT1 in order to suggest causation, but this is not our intention with the bioinformatic analysis. We merely wanted to highlight that association between these genes exist in clinical setting, and the 1/20th of breast tumours in a large dataset is a clinically important subset, underlining the importance of the functional findings in the paper.

I think that it would be good to have an informaticist skilled in the analysis of gene correlations in tumors review all of the gene analyses in this report. I also have concerns about the way that other aspects of the informatics analysis were performed."

The bioinformaticians included in the authors' list have re-examined all the analysis presented in the paper, and apart from minor modifications (definition of samples overexpressing the PSAT1 and IKBKE genes), agreed on all the results presented previously.

Dear Katuscia,

Thank you for the re-submission of your manuscript to EMBO reports. Your manuscript has now been seen by a member of our advisory board, and by an external expert in statistics and bioinformatics (referee #3 - referee #1 was the one from TEJ, referee #2 is our board member), working in a cancer context.

Our advisor states:

'Based on the severe criticism of the referee regarding lacking validity of the statistical data, his/her further points, and comparing the rebuttal and revised final manuscript, I cannot give a green light for this manuscript. I found the data on PSAT1 phosphorylation and the further experimentally unsubstantiated role of IKK ϵ -associated TBK1 towards PSAT1 still unacceptable. A chapter describing that PSAT1 phosphorylation has no function is also a distractive part in the paper.'

There was only one referee for the EMBO Reports submission and he/she brings up hefty concerns regarding the statistical significance of the conclusions regarding the breast cancer data analysis. The statistical analysis tools are outside of my area of expertise. Therefore, I suggest engaging an additional referee with expertise in statistics and bioinformatics analysis. Without positive feedback from such a person, I cannot support acceptance of the current version.'

The manuscript was thus sent to a statistics/bioinformatics expert to get his/her feedback (which took much longer than expected). Please find his/her report below.

As you will see, the referee states that the present analysis does not support the major claim, i.e. that IKK ϵ and PSAT1 expression are positively correlated in a subgroup of tumors. As the report is below, I will not further detail this here.

Given these substantial remaining concerns, the fact that you already had more than one opportunity to revise the study, and also considering that both referees do not provide strong support for the publication of the study in EMBO reports and are not convinced by the current set of data, I am afraid that we cannot offer to publish the manuscript.

I am sorry that this final decision emerges as the outcome of a lengthy (re-)review process. However, I do not think that it does make much sense to continue with this manuscript here, also taking into account that we usually only allow one round of major revisions. I nevertheless hope that the comments will be helpful to revise and re-organize the study, and to give it a fresh try at another journal.

Yours sincerely,

Achim Breiling
Editor
EMBO Reports

Referee #3:

I was asked to assess the statistical / bioinformatics part of the analysis of the paper by Xu et al. I am not an expert in cancer metabolism or proteomics and will thus not comment on the general suitability of the manuscripts for EMBO Reports, or on the biological impact.

General comments:

The ms contains multiple tests for difference in means between various measurements (e.g. between ion intensities of $^{13}\text{C}_6$ labeled glucose, Fig 1). From the bar plots in the figures it is unclear how the data points are distributed, what test was used to assess difference in means and what the p-value is. I would like to see a dotplot-like presentation such as provided by the authors e.g. in Fig 7i. The test used and p-value obtained should be mentioned in the main text where the result is first presented.

In Fig 3a, variance explained by the principal components should be given.

Specific comments:

The main controversial point seems to be about the correlation analyses conducted (Fig 7 F,G,H,J and EV7 A,B,G). From the data presented it is clear that there is a consistent difference in gene expression between tumour types (in particular between UFb and LF/UF). However, the correlation values provided in my eyes are likely to only reflect this difference between subtypes. Within each group, I do not see a strong correlation between either PHGDH, PSAT1 or PSPH and IKBKE (Fig 7F-H). From the authors comments in the rebuttal email I take that the 'first 50 of ranking' are of importance. Unfortunately this subset is not marked in the figure, but if it also contains all tumour subtypes, the observed correlation likely again reflects difference in expression levels between subtypes, not co-expression of the studied genes. This seems to be confirmed by EV7A, where after discretization the chi2 test shows a strong dependence between the groups.

If the authors want to convincingly show a co-expression of the genes (overall or in the 'first 50 of ranking'), I would suggest a mixed model analysis that accounts for overall differences in expression levels between the tumour types as a random effect. This might be a way of solving this debate and verifying that the correlations hold independent of the grouping of samples. Alternatively, independent correlation tests within each tumour subtype, as suggested by the referee, would be able to verify or refute the authors' claims.

In its current form, the analysis does not convince me of the claim (quoting from rebuttal): "Our aim with the bioinformatic analysis of IKBKE and PSAT1 expression was to simply show that there is a subgroup of tumours, where IKBKE and PSAT1 are co-expressed", as the correlation that is being picked up by the linear model is likely simply due to the subtypes, not due to actual co-expression.

** As a service to authors, EMBO Press provides authors with the ability to transfer a manuscript that one journal cannot offer to publish to another journal, without the author having to upload the manuscript data again. To transfer your manuscript to another EMBO Press journal using this service, please click on
Link Not Available

Dear Katuscia,

Thank you for the re-submission of your research manuscript to EMBO reports. We have now received reports from the two referees that were asked to evaluate your study, which can be found at the end of this email.

As you will see, both referees think that the findings are of interest, but they also have several comments and suggestions, we ask you to address in a final revised version of the manuscript. Please also provide a point-by-point-response addressing the remaining points by the referees.

Further, I have these editorial requests:

- We require individual production quality figure files as .eps, .tif, .jpg (one file per figure), of main figures and EV figures. Please upload these as separate, individual files upon re-submission.

For more details please refer to our guide to authors:

Please also format the methods section according these guidelines.

See also our guide for figure preparation:

- We also require that primary datasets produced in this study (e.g. RNA-seq, ChIP-seq and array data) are deposited in an appropriate public database. This is now mandatory (like the COI statement). If no primary datasets have been deposited in any database, please state this in this section (e.g. 'No primary datasets have been generated and deposited').

The accession numbers and database should be listed in a formal "Data Availability " section (placed after Materials & Methods) that follows the model below. Please note that the Data Availability Section is restricted to new primary data that are part of this study.

Data availability

- Please also provide an updated author checklist, which you can download from our author guidelines (<https://www.embopress.org/page/journal/14693178/authorguide>). Please insert page numbers in the checklist to indicate where the requested information can be found in the manuscript. Please make sure that all relevant fields are filled in.

- Please note that we are changing our reference style. Please format the references to our new reference style. See:

- There are two tables, which are large datasets. Please upload these as Datasets and name them Dataset EV1, and Dataset EV2, and use this nomenclature for the call-outs in the manuscript. Please provide a legend for these datasets on the first TAB of the excel files. Finally, please remove their legends from the main manuscript text.

- the abstract is slightly too long. Please shorten it to not more than 175 words.

- We strongly encourage the publication of original source data (in particular of Western blots) with the aim of making primary data more accessible and transparent to the reader. The source data will be published in a separate source data file online along with the accepted manuscript and will be linked to the relevant figure. Please submit the source data (scans of entire gels or blots, data points of graphs in an excel sheet, additional images, etc.) of your key experiments together with the revised manuscript. Please include size markers for scans of entire gels, label the scans with figure and panel number, and send one PDF file per figure.

- Regarding data quantification and statistics, can you please check again that, where applicable, the number "n" for how many independent experiments (biological replicates) were performed, the bars and error bars (e.g. SEM, SD) and the test used to calculate p-values is indicated in the respective figure legends. Please provide statistical testing where applicable, and also add a paragraph detailing this to the methods section. See:

<http://www.embopress.org/page/journal/14693178/authorguide#statisticalanalysis>

- Finally, please find attached a word file of the manuscript text (provided by our publisher) with changes we ask you to include in your final manuscript text, and some queries, we ask you to address. This file was prepared for the V2 version of the paper! Please address all queries that also apply to the final version of the paper. In particular, please indicate for all replicates if these were technical or biological. Please provide your final manuscript file with track changes, in order that we can see the modifications done.

In addition I would need from you:

- a short, two-sentence summary of the manuscript

- two to three bullet points highlighting the key findings of your study
- a schematic summary figure (in jpeg or tiff format with the exact width of 550 pixels and a height of not more than 400 pixels) that can be used as a visual synopsis on our website.

Kind regards,

Achim

Achim Breiling
Editor
EMBO Reports

Referee #1:

Comment 1

The authors report that

"The increased intracellular level of serine was a consequence of increased biosynthesis as we observed a significant increase in the level of $^{13}\text{C}_6$ -glucose-derived serine (m+3 isotopologue), suggesting that IKK ϵ positively regulates the SBP (Fig. 1C)."

"Using $^{15}\text{N}_2$ -glutamine labelling, we confirmed increased levels of nitrogen labelling of serine (m+1) in IKK ϵ expressing cells (see Fig. 1C and Table 1), consistent with an increase in PSAT1 transamination activity, corroborating that serine biosynthesis was activated by IKK ϵ ."

"Finally, the fraction of ^{15}N labelled serine derived from glutamine was also increased, indicating higher glutamine usage in serine biosynthesis as nitrogen source (Fig. 1G)."

However, that data is not sufficient to arrive to those conclusions. Inferring metabolic fluxes from metabolite concentrations is a challenging problem. An increase in the concentration of a metabolite can be achieved by an increase of production but also by a decrease of consumption, or a combination of both. The authors are claiming that the increase in serine levels can be taken as evidence for an increase in serine synthesis. However, serine levels could be also increased by a decrease in serine consumption. The latter is relevant in this context because mitochondria catabolize serine coupled to the rate of oxidative phosphorylation. Since IKK ϵ inhibits mitochondrial respiration I would also expect that it inhibits the mitochondrial catabolism of serine, providing an alternative hypothesis for the observed increase in serine levels.

One way to tackle this problem is to conduct a metabolic flux analysis around serine, using as input the [U- ^{13}C]-glucose tracing data. The explanation of this methodology is provided in Tedeschi et al [PMID: 26023330], Additional file 1, Section 2.3. The equations (2.3.12) and (2.3.13) provide a straightforward calculation to estimate the rate of serine synthesis from glucose and from glycine, using as input the serine and glycine isotopologue fractions and the rate of serine uptake. To apply

this methodology the authors should measure the rate of serine uptake in all their conditions. Reporting the rates of serine uptake in the conditions of Figure 1 and 2 will also help to better understand the changes in serine metabolism.

Comment 2

The IKKepsilon dependent inhibition of mitochondrial activity could increase ROS levels, which could then lead to the induction of Nrf2 and as a consequence an enhancement of ATF4 activity. The heatmap in Fig. 1B suggest that there is a reduction on glutathione levels upon the induction of IKKepsilon expression. The data in Figure 2D suggest that there is an increase on GSH levels upon knockdown of IKKepsilon expression, although the opposite happens in T47D cells (Fig. 2C). Glutamate, a precursor of glutathione synthesis, is also reduced by IKKepsilon induction (Fig. 1B) and increased by IKKepsilon knockdown (Fig. 2CD). The reduction in glutamate levels upon IKKepsilon induction may also cause a reduction in the uptake of cystine coupled to glutamate release, and cystine is another precursor of glutathione synthesis.

Based on this discussion I request measurements of ROS levels and Nrf2 levels to determine whether they are induced by IKKepsilon. If there is an increase in ROS levels upon induction of IKKepsilon then the authors should test whether the induction of ATF4 is reduced by an antioxidant like NAC.

Comment 3

It is not clear from the reported data whether the increase in PSAT1 expression and the putative increase in serine synthesis (to be tested as discussed above) are a bystander, or are they required to support the phenotypic changes associated with IKKepsilon induction. The authors should knockdown the expression of PSAT1 in the context of IKKepsilon induction in HEK-293 cells and investigate whether that affects proliferation and/or viability.

Comment 4

I am not convinced that IKKepsilon could be a major determinant of PSAT1 expression in breast tumours. Myc also induces the expression of serine synthesis genes and it is commonly activated in basal breast cancers. The authors should infer the activity of NFKB1, ATF4 and Myc using gene expression signatures. MSigDB (<https://www.gsea-msigdb.org/gsea/msigdb/index.jsp>) contains gene signatures listing the targets of NFKB1, ATF4 and Myc that can be used to infer their transcriptional activity using as input gene expression data (e.g., IGARASHI_ATF4_TARGETS_UP, RASHI_NFKB1_TARGETS, DANG_MYC_TARGETS_UP).

Comment 5, minor

Figure 2E. Can the authors explain how Glycine+1 is generated from ¹⁵N₂-Glutamine? Is this a surrogate of reverse glycine cleavage?

Referee #2:

This is a very interesting manuscript. The paper is clearly written and the figures are very good. I think the topic is relevant too. The conclusions are supported by the data.

Small comment:

Statistical analysis in the graph. All the graph in the paper has one (*) but in the figure legend there is the correct statistical analysis. Using asterisks ("*") to denote statistical significance in a graph, is the easiest way to show significance level between two groups samples.

For example:

* for difference less than $P < 0.01$,

** for difference less than $P < 0.001$,

*** for difference less than $P < 0.0001$.

Re: EMBOR-2019-48260V4-Q – Referees point by point response

Dear Achim,

Thanks for having sent us the comments of Referees.

With regards of the comments of Referee #1, we have provided extra additional experiments to address the concerns raised. We believe that, as detailed below, no further data will be required to support our conclusions and where important points were raised we modified the manuscript accordingly.

With regards of the comments of Referee#2. We are pleased that the referee found our work interesting and was supportive of publication and have addressed the his/her minor comment and added stars in the figures according to the p-value.

Referre#1

Comment 1

..."However, that data is not sufficient to arrive to those conclusions. Inferring metabolic fluxes from metabolite concentrations is a challenging problem. An increase in the concentration of a metabolite can be achieved by an increase of production but also by a decrease of consumption, or a combination of both. The authors are claiming that the increase in serine levels can be taken as evidence for an increase in serine synthesis. However, serine levels could be also increased by a decrease in serine consumption. The latter is relevant in this context because mitochondria catabolize serine coupled to the rate of oxidative phosphorylation. Since IKK ϵ inhibits mitochondrial respiration I would also expect that it inhibits the mitochondrial catabolism of serine, providing an alternative hypothesis for the observed increase in serine levels...."

We agree with the referee that "Inferring metabolic fluxes from metabolite concentrations is a challenging problem" and indeed we have not made any claim about metabolic fluxes in our manuscript. Although we cannot exclude that the increase in serine is due to a defect in its catabolism, our data clearly show that IKK ϵ induces the upregulation of the SBP enzymes (Fig. 4-5) and the parallel increase in the levels of glucose- and glutamine-derived serine when using independent tracers (Fig. 1-2). Thus, the experiment suggested by the referee would add information on serine catabolism, which is however not the main focus of our work. Moreover, we tested the level of expression of SHMT2 (the main mitochondrial enzyme responsible for serine catabolism in the mitochondria) in our model systems and could not detect any differences. We added these data to the manuscript as a Fig. EV2B,C and also to rephrase the text as indicated below.

*"The increased intracellular level of serine was **likely** a consequence of increased biosynthesis as we*

observed a significant increase in the level of $^{13}\text{C}_6$ -glucose-derived serine ($m+3$ isotopologue), suggesting that IKK ϵ positively regulates the SBP (Fig. 1C)."

"Using $^{15}\text{N}_2$ -glutamine labelling, we confirmed increased levels of nitrogen labelling of serine ($m+1$) in IKK ϵ expressing cells (see Fig. 1C and Table 1), consistent with an increase in PSAT1 transamination activity, ~~corroborating~~ **supporting our hypothesis** that serine biosynthesis was activated by IKK ϵ ."

"Finally, the fraction of ^{15}N labelled serine derived from glutamine was also increased, indicating higher glutamine usage in serine biosynthesis as nitrogen source (Fig. 1G)."

Moreover, our hypothesis that modulation of pyruvate uptake in the mitochondria regulates serine biosynthesis has been recently confirmed by (Baksh *et al*, 2020) and this paper is now cited in the discussion.

Comment 2

... "Based on this discussion I request measurements of ROS levels and Nrf2 levels to determine whether they are induced by IKKepsilon. If there is an increase in ROS levels upon induction of IKKepsilon then the authors should test whether the induction of ATF4 is reduced by an antioxidant like NAC."

This is a fair point, and previously we have considered assessing ROS in our system. It has been reported that IKK ϵ regulates ROS production (Chai *et al*, 2020) and we cannot exclude the involvement of ROS/NRF2 in the cascade, thus this point has now been added to the discussion. However, given that NRF2 is considered upstream to ATF4 (DeNicola *et al*, 2015), these experiments would not alter the model that we propose, but would just add an additional level of complexity that would require substantial evidence to be corroborated.

Comment 3

It is not clear from the reported data whether the increase in PSAT1 expression and the putative increase in serine synthesis (to be tested as discussed above) are a bystander, or are they required to support the phenotypic changes associated with IKKepsilon induction. The authors should knockdown the expression of PSAT1 in the context of IKKepsilon induction in HEK-293 cells and investigate whether that affects proliferation and/or viability.

In this comment the referee asks to clarify the functional consequences of IKK ϵ -mediated upregulation of the SBP, which we are extensively investigating. [Figures for referees not shown.]

Moreover, we added further data to the manuscript showing that IKK ϵ -mediated inhibition of the mitochondria is not dependent on PSAT1 (Fig. EV3A) and that PSAT1 siRNA itself does not affect the OCR in our panel of breast cancer cell lines (Fig. EV3B). These data indicate that IKK ϵ regulates the SBP via mitochondria and excludes that the effect on the mitochondria is mediated via modulation of the SBP.

Comment 4

I am not convinced that IKKepsilon could be a major determinant of PSAT1 expression in breast tumours. Myc also induces the expression of serine synthesis genes and it is commonly activated in basal breast cancers. The authors should infer the activity of NFKB1, ATF4 and Myc using gene expression signatures. MSigDB (<https://www.gsea-msigdb.org/gsea/msigdb/index.jsp>) contains gene signatures listing the targets of NFKB1, ATF4 and Myc that can be used to infer their transcriptional activity using as input gene expression data (e.g., IGARASHI_ATF4_TARGETS_UP, RASHI_NFKB1_TARGETS, DANG_MYC_TARGETS_UP).

We would be happy to perform the suggested analysis. However, there are a few points we would like to highlight:

1. We have acknowledged in the manuscript that *“Importantly, while PSAT1 is overexpressed in almost all ER negative samples (378 out of 435), only 79 samples overexpress IKBKE, indicating that PSAT1 is regulated by multiple inputs”*. Thus, even if Myc and ATF4 are involved in the regulation of PSAT1 independently of IKK ϵ , our conclusion that in a subset of tumours the functionally described mechanism can exist, remains still valid.
2. Performing the analysis suggested by Referee #1 will allow us to possibly establish a correlation between the level of expression of IKK ϵ and PSAT1 with Myc, NFKB1 or ATF4 gene expression signature. However, this correlation will have to be functionally tested, which is what we have already done in the paper, demonstrating a role for ATF4 (Figure 4) and excluding a role for NF κ B (Figure S2). We have now also added functional data for Myc (Fig. EV2A) showing that the level of expression of

Myc in our HEK model system does not change upon induction of IKK ϵ , whilst on the contrary ATF4 is strongly induced (Figure 4A).

Comment 5, minor

Figure 2E. Can the authors explain how Glycine+1 is generated from 15N2-Glutamine? Is this a surrogate of reverse glycine cleavage?

We believe this comment has been raised because of a misunderstanding caused by our label of Figure 2E – left panel, which did not clearly state that the “m+1” referred to nitrogen, and not carbon. We apologise for the confusion and have modified the labelling.

In conclusion, we would like to thank the referee for his/her comments that lead us to add new data (i.e. the ones relative to Myc and SHMT2) and discuss important points (such as NRF2) that made our hypothesis stronger.

Referee #2

Small comment:

Statistical analysis in the graph. All the graph in the paper has one (*) but in the figure legend there is the correct statistical analysis. Using asterisks ("*") to denote statistical significance in a graph, is the easiest way to show significance level between two groups samples.

For example:

- * for difference less than $P < 0.01$,
- ** for difference less than $P < 0.001$,
- *** for difference less than $P < 0.0001$.

We have addressed this point and statistics are now shown on graphs as suggested by Referee #2.

Katiuscia Bianchi
Barts Cancer Institute, Queen Mary University of London
Charterhouse Square
London EC1M 6BQ
United Kingdom

Dear Dr. Bianchi,

I am very pleased to accept your manuscript for publication in the next available issue of EMBO reports. Thank you for your contribution to our journal.

At the end of this email I include important information about how to proceed. Please ensure that you take the time to read the information and complete and return the necessary forms to allow us to publish your manuscript as quickly as possible.

As part of the EMBO publication's Transparent Editorial Process, EMBO reports publishes online a Review Process File to accompany accepted manuscripts. As you are aware, this File will be published in conjunction with your paper and will include the referee reports, your point-by-point response and all pertinent correspondence relating to the manuscript.

If you do NOT want this File to be published, please inform the editorial office within 2 days, if you have not done so already, otherwise the File will be published by default [contact: emboreports@embo.org]. If you do opt out, the Review Process File link will point to the following statement: "No Review Process File is available with this article, as the authors have chosen not to make the review process public in this case."

Should you be planning a Press Release on your article, please get in contact with emboreports@wiley.com as early as possible, in order to coordinate publication and release dates.

Thank you again for your contribution to EMBO reports and congratulations on a successful publication. Please consider us again in the future for your most exciting work.

Yours sincerely,

Achim Breiling
Editor
EMBO Reports

THINGS TO DO NOW:

You will receive proofs by e-mail approximately 2-3 weeks after all relevant files have been sent to our Production Office; you should return your corrections within 2 days of receiving the proofs.

Please inform us if there is likely to be any difficulty in reaching you at the above address at that time. Failure to meet our deadlines may result in a delay of publication, or publication without your corrections.

All further communications concerning your paper should quote reference number EMBOR-2019-48260V5 and be addressed to emboreports@wiley.com.

Should you be planning a Press Release on your article, please get in contact with emboreports@wiley.com as early as possible, in order to coordinate publication and release dates.

YOU MUST COMPLETE ALL CELLS WITH A PINK BACKGROUND ↓
PLEASE NOTE THAT THIS CHECKLIST WILL BE PUBLISHED ALONGSIDE YOUR PAPER

Corresponding Author Name: Katuscia Bianchi
Journal Submitted to: EMBO Reports
Manuscript Number: EMBOR-2019-48260V3